# Using mortuary and burial data to place COVID-19 in Lusaka, Zambia within a global context

Richard J. Sheppard[1], Oliver J. Watson [1,2], Rachel Pieciak[3], James Lungu[4], Geoffrey Kwenda[5], Crispin Moyo[4], Stephen Longa Chanda [6], Gregory Barnsley [1], Nicholas F. Brazeau[1], Ines C. G. Gerard-Ursin[1], Daniela Olivera Mesa [1], Charles Whittaker [1], Simon Gregson[1,7], Lucy C. Okell [1], Azra C. Ghani [1], William B. MacLeod[3], Emanuele Del Fava [8,9], Alessia Melegaro [8,10], Jonas Z. Hines[11], Lloyd B. Mulenga[12], Patrick G. T. Walker[1,13] ✉, Lawrence Mwananyanda[3,4,13] & Christopher J. Gill[3,13]

Reported COVID-19 cases and associated mortality remain low in many sub-Saharan countries relative to global averages, but true impact is difficult to estimate given limitations around surveillance and mortality registration. In Lusaka, Zambia, burial registration and SARS-CoV-2 prevalence data during 2020 allow estimation of excess mortality and transmission. Relative to pre-pandemic patterns, we estimate age-dependent mortality increases, totalling 3212 excess deaths (95% CrI: 2104–4591), representing an 18.5% (95% CrI: 13.0–25.2%) increase relative to pre-pandemic levels. Using a dynamical model-based inferential framework, we find that these mortality patterns and SARS-CoV-2 prevalence data are in agreement with established COVID-19 severity estimates. Our results support hypotheses that COVID-19 impact in Lusaka during 2020 was consistent with COVID-19 epidemics elsewhere, without requiring exceptional explanations for low reported figures. For more equitable decision-making during future pandemics, barriers to ascertaining attributable mortality in low-income settings must be addressed and factored into discourse around reported impact differences.

Following the emergence of COVID-19 in 2019, SARS-CoV-2 has spread across the world, causing the highest level of social and economic disruption due to an infectious disease in living memory[1]. As of April 24th 2023, confirmed global cases and deaths total 763,740,140 and 6,908,554 respectively[2]. This recorded impact has occurred despite the implementation of often stringent non-pharmaceutical interventions (NPIs) that frequently proved capable of controlling the epidemic while in place[3,4]. The subsequent development of numerous vaccines

[1]MRC Centre for Global Infectious Disease Analysis, Department of Infectious Disease Epidemiology, Imperial College, London, UK. [2]Department of Infectious Disease Epidemiology, Faculty of Epidemiology and Population Health, London School of Hygiene and Tropical Medicine, London, UK. [3]Department of Global Health, Boston University School of Public Health, Boston, MA, USA. [4]Avencion Limited, Lusaka, Zambia. [5]Department of Biomedical Sciences, School of Health Sciences, University of Zambia, Lusaka, Zambia. [6]Zambia National Public Health Institute, Lusaka, Zambia. [7]Manicaland Centre for Public Health Research, Biomedical Research and Training Institute, Harare, Zimbabwe. [8]Carlo F. Dondena Centre for Research on Social Dynamics and Public Policy, Bocconi University, Milan, Italy. [9]Max Planck Institute for Demographic Research, Rostock, Germany. [10]Department of Social and Political Science, Bocconi University, Milano, Italy. [11]Centers for Disease Control and Prevention, Lusaka, Zambia. [12]Zambia Ministry of Health, Lusaka, Zambia. [13]These authors contributed equally: Patrick G. T. Walker, Lawrence Mwananyanda, Christopher J. Gill. ✉e-mail: patrick.walker06@imperial.ac.uk

has also mitigated the potential impact of COVID-19 through decreasing the rates of severe disease in all known genetic variants[5–8] and, to a lesser-extent, lowering transmission[9,10].

Relative to a global average of 8128.8 reported cases and 84.6 reported deaths per 100,000 (as of April 24th 2023), reported COVID-19 impact in Africa remains notably low (884.7 cases and 15.6 deaths per 100,000)[2]. Common hypotheses for a perceived low impact of COVID-19 in Africa include a younger population structure[11–21], protection through exposure to pre-existing coronaviruses and other endemic diseases[11,13,15–17,20,22], climate-related protections[14–17,19,20,23], effective use of NPIs developed from disease-control expertise[11,15,17,20,24], and genetic factors[13,16,17,25]. Some hypotheses (low population density and climate-associated outdoor lifestyles[14–20,23]) suggest a landscape of reduced disease transmission, whilst others assume high transmission with an associated implication of lower severity (e.g., protection through existing coronavirus exposure). However, it is widely acknowledged that recorded cases and deaths substantially under-estimate the total number of SARS-CoV-2 infections and COVID-19 attributable deaths in many settings, corresponding with regional testing capacities[26–30]. This raises a fundamental question: once ascertainment biases are accounted for, to what extent are any of these alternative hypotheses needed? Understanding whether there was any so-called "Africa paradox"[12,16,17] (the perception of lower COVID-19 impacts across many African countries compared with expectations) and if so, which, if any, of these hypotheses have justifiable basis can help ensure that the correct conclusions and lessons are learned from the pandemic in Africa.

While an increasing amount of evidence, as of late 2022, suggests that SARS-CoV-2 has spread widely in many African countries[30,31], similar data from 2020, when the concept of low impact in Africa gained traction, remain exceptionally scarce. Of 134 high-quality large-scale community-based seroprevalence surveys from 2020, only five were from African countries[32]. Meanwhile, capturing trends in excess mortality (as a proxy for COVID-19 impact) in Africa during the pandemic is challenging due to a lack of robust civil and vital registration systems in many countries. In a recent global study of excess mortality, the World Health Organisation (WHO) found that 41 of 47 of the WHO's Africa region countries had no data suitable for inclusion, resulting in a wide uncertainty interval for the Africa region[33]. Furthermore, when excess mortality trends can be well-quantified, it is critical to identify where changes in overall pandemic mortality have indirect (e.g., reduced traffic accidents due to NPIs) or direct causes (i.e., infection-associated disease)[34,35]. Further delineating differences in direct impact between regions due to *spread* (i.e., the percentage of population infection) and severity (often summarised as the infection-fatality-ratio, IFR) is then critical to understanding the need for and impact of control measures. IFR estimation in African countries has been challenging due to difficulties in the ascertainment of SARS-CoV-2 infections and COVID-19 deaths, resulting in estimates that vary substantially as different mortality estimates are used[19,29,36].

Research conducted in Zambia generated important insights into the true impact of the pandemic in Africa during 2020. In July 2020, during the first pandemic wave in Zambia, a population-based survey found that 2.1% (95% CI: 1.1–3.1%) of the population had evidence of previous infection based on an IgG serological assay while 7.6% (95% CI: 4.7–10.6%) tested positive for active infection by polymerase chain reaction (PCR)[37]. These results represent a 92-fold increase in infection rates relative to reported case number within the country at the time[37]. Meanwhile, post-mortem sampling of the mortuary at the University Teaching Hospital (UTH), the largest morgue in Lusaka, the capital of Zambia, found 15% of deaths tested positive by PCR (CT < 40) during June-October 2020[27], proportions that were exceeded during subsequent pandemic waves in 2021[38]. All-cause burial registrations were also collected from official registries[39] by the post-mortem sampling team, dating from mid-2017 to mid-2021. These data provide opportunity for inference of SARS-CoV-2 spread and COVID-19 severity in Lusaka that can be compared with estimates from elsewhere.

In this study we build upon this combined research effort to ask: "Is there evidence that age-patterns of COVID-19 severity in Lusaka were substantially different from patterns in other countries during the first wave of the pandemic?". To answer this, we place existing WHO excess mortality estimates during the pandemic in Zambia in a global context, explicitly accounting for the protective effect of its relatively young population. We then develop a statistical framework to obtain estimates of pandemic impact upon Lusaka mortality patterns, focusing upon the burial registration age-distribution during 2020–2021. We combine these mortality estimates with data from the Lusaka population-based PCR- and sero-survey and UTH post-mortem PCR testing to assess whether patterns of excess mortality reflect likely patterns of spread. Finally, we use this approach to provide inference on key parameters such as the reproduction number, cumulative attack rate and IFR within Lusaka during 2020, comparing these with estimates throughout the world (Fig. 1). We find evidence of substantial COVID-19 impact on excess mortality during epidemic waves, comparable to that seen elsewhere, and that previously established COVID-19 severity estimates are in statistical agreement with observed mortality and SARS-CoV-2 prevalence patterns in the city. These results suggest that low reported COVID-19 cases and deaths in Lusaka are not reflective of true patterns of spread and severity. Limitations in the ascertainment of disease burden in low-income countries must therefore be incorporated into pandemic planning in order to improve decision-making and health outcomes in future pandemics.

## Results

### WHO 2020 excess mortality estimates in Zambia

The WHO's mean per-capita excess mortality estimate of 290 deaths per million in Zambia during 2020 ranks 26th highest of the 47 WHO Africa region countries (Fig. 2a). The estimate is similar to the equivalent Africa region estimate of 320 deaths per million, which ranks 5th highest, above only the Western Pacific region, where estimated 2020 excess mortality was negative. Zambia's uncertainty estimate is wide (and of similar magnitude to 40 other African countries in which the WHO was unable to identify adequate mortality data) allowing only limited comparative analysis with the rest of the world. Zambia's upper bound of 820 deaths per million excludes Algeria and South Africa's lower bound estimates (both of which had well-documented large-scale epidemics[40]) and of the Europe and the Americas WHO regions. Meanwhile, its lower bound of -230 deaths per million exceeds the upper bound of estimates for Seychelles, Mauritius, Kenya, Togo and encompasses the region-level estimates for the Western Pacific region. Seychelles and Mauritius, two more developed island nations in the Africa region, are known to have experienced lower-than-typical rates of overall 2020 mortality (likely driven by suppression measures involving stringent border controls[40,41]). Aside from these comparisons, little can be said about the actual excess mortality in any country with similarly wide confidence intervals.

To help place these estimates in the context of differing underlying demography, we first weighted globally-derived estimates of age-specific IFR[42] by population age-structure. According to this measure of demographic vulnerability to severe disease, Zambia ranked 2nd lowest in both Africa and the world, with only Uganda's demographic structure producing a lower estimate (Fig. 2b). From these estimates, we then calculated a measure that standardises excess mortality by average protection from (or vulnerability to) severe disease upon infection that comes from population age structure. This measure, the "demographic vulnerability-weighted impact" (DVWI), is defined explicitly as the cumulative attack rate required to match estimates of excess mortality, assuming direct COVID-19 causation and age-specific IFR from Brazeau et al.[42], with even spread of infection by age within the population. It is important to note that indirect pandemic

# SARS-CoV-2 transmission and severity

**Fig. 1 | Inferential Framework.** Data sources and other inputs are denoted by purple boxes, methodological steps are shown in orange boxes, while results and other outputs are shown in teal boxes. **A** Age-stratified burial registration data are used to **B** quantify the shift in registration age-patterns throughout the pandemic, relative to those observed in 2018–2019. These are then converted into **C** excess mortality estimates during 2020 until June 2021, making several assumptions (subjected to various sensitivity analyses), in particular that registration rate changes in children (mirrored by similar patterns in adolescents and young adults) during the pandemic are a guide to underlying registration and mortality patterns. These estimates are combined with: **D** weekly post-mortem polymerase chain reaction (PCR) prevalence data from the largest mortuary in Lusaka during June-October 2020; **E** population-based PCR prevalence and seroprevalence survey data from July 2020; **F** demography information and likely social-contact structure within Lusaka. These inputs are then used to **G** fit an age-structured SARS-CoV-2 transmission model using Markov chain Monte Carlo for **H** a given infection-fatality ratio (IFR) pattern by age to **I** infer transmission trends over time, **J** extrapolate patterns of spread throughout the first pandemic wave in Lusaka and **K** obtain the posterior likelihood of observing the patterns in **B**, **D**, and **E** conditional on the assumed IFR pattern by age.

consequences also impact excess mortality. Moreover, our estimates are based upon infection-fatality patterns during the pandemic's first wave, largely using data from high-income settings with good care access and standards, relative to global averages. Consequently, this measure is not designed to provide insight into excess mortality causes, but places such estimates within the context of the population's vulnerability to direct infection consequences at the beginning of the pandemic. It is, therefore, plausible that countries can have DVWI>1 due to any combination of: (i) high indirect pandemic impact; (ii) greater disease severity, due to health-care limitations, SARS-CoV-2 variants of greater severity or any other factor not accounted for which contribute to higher IFRs by age than those used in our analysis; (iii) substantial burden associated with reinfection.

Our estimates (Fig. 2c) highlight that, once demographic effects are removed, uncertainty in existing WHO estimates for Zambia (DVWI = 0.251, 95% CI: 0–0.710) permit very few conclusions about the differential impact of the disease relative to global patterns. We find that estimates for Zambia are no longer comparatively lower than the worst impacted Africa region countries such as South Africa (DVWI = 0.294, 95% CI: 0.265–0.320) or Algeria (DVWI = 0.417, 95% CI:

0.397–0.440), nor lower than the worst impacted WHO regions such as Europe (DVWI = 0.171, 95% CI: 0.167–0.176) and the Americas (DVWI = 0.240, 95% CI: 0.233–0.248).

## Burial registration patterns in Lusaka

Figure 3 shows officially reported COVID-19 deaths for Lusaka Province[43] and burial registration data[39] during January 2018–June 2021. NPIs restricting international travel, closing educational and social establishments, limiting gatherings to 50, and encouraging social distancing, hand sanitation and mask wearing were implemented in Zambia in late March 2020[44–46], concurrently with the country's first reported COVID-19 cases 19th March 2020[47]. Very few confirmed COVID-19 deaths occurred whilst these measures were maintained (Fig. 3a). However, a first wave of 223 confirmed COVID-19 provincial deaths occurred during June-October 2020, following a gradual easing of initial restrictions during 24th April–25th June, 2020 (reopening recreational venues, airports, and educational institutions for older students. Social distancing, mask wearing and gathering limitations were maintained throughout)[48–51]. Two subsequent waves of 273 and 182 confirmed deaths occurred during January-March 2021

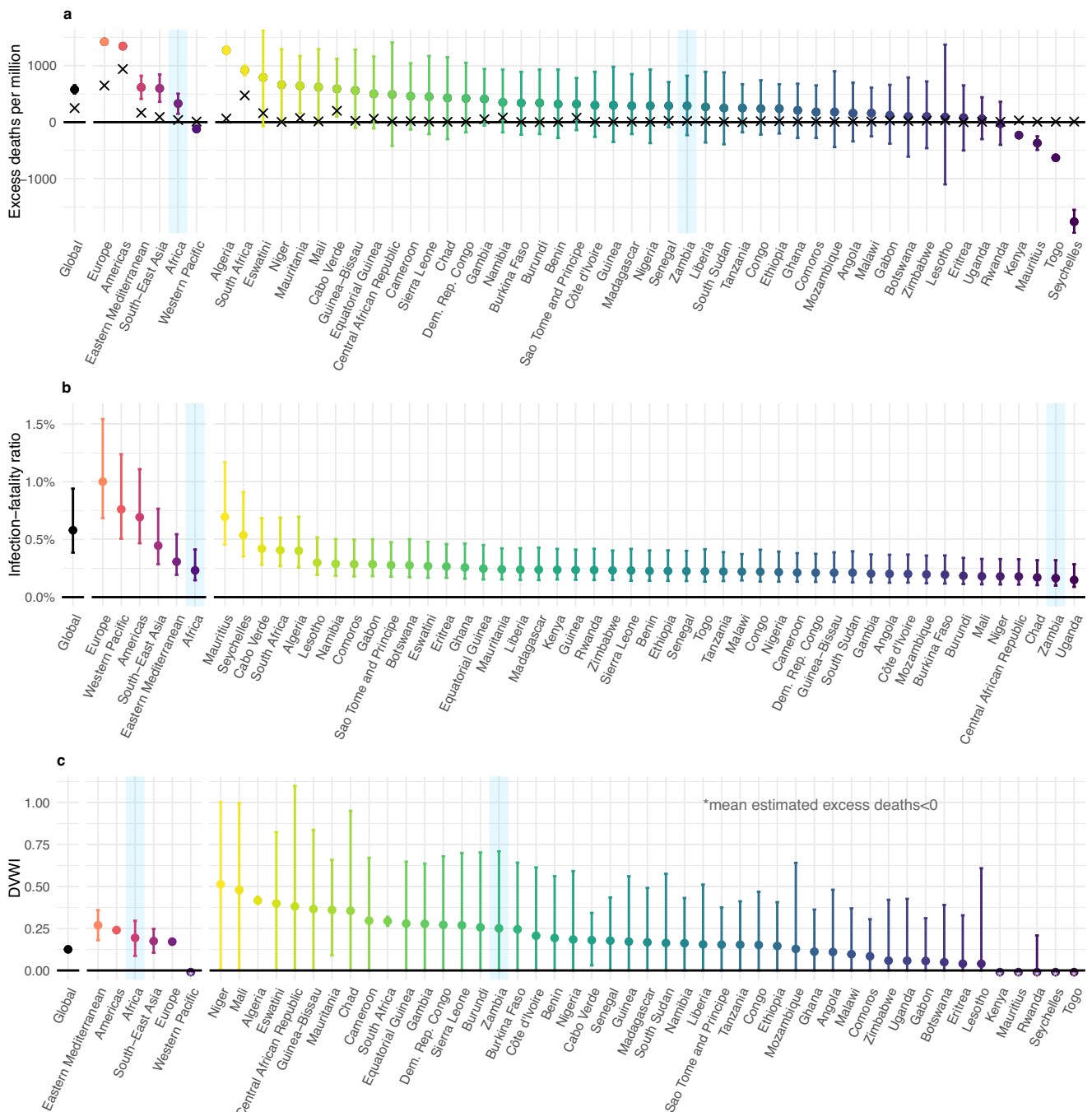

**Fig. 2 | Global estimates of excess mortality relative to patterns of demographic vulnerability.** Figure shows **a** World Health Organisation (WHO) estimates of excess mortality per million people in 2020. Points show mean and lines 95% confidence intervals from 1000 samples. Crosses show confirmed COVID-19 mortality per million people in 2020. **b** Estimates of region-level IFR calculated using age-specific IFR estimates from Brazeau et al.[42] weighted by region population age-distribution (i.e., assuming infection equally distributed across the population). Points show median IFR and lines 95% credible intervals from 1000 draws of the joint posterior of the IFR by age curve. **c** Estimated demographic-vulnerability-weighted impact (DVWI), defined as the cumulative attack rate, spread uniformly by age, required to achieve a level of direct COVID-19 mortality matching the excess mortality in **a** assuming the posterior median IFR from **b**. Points and lines show median with 95% confidence intervals corresponding with 1000 draws from excess mortality estimates in **a**. All panels highlight in blue estimates for the WHO Africa region and Zambia for ease of identification.

and from April to June, 2021 respectively. Post-mortem sampling[38] and GISAID data[52] show these waves were associated with the Beta and Delta variants.

Although burial registration is legally required in Zambia, deaths are not registered comprehensively, particularly when a substantial proportion occur in the community[53,54]. Overall, 14,665 and 14,992 Lusakan deaths were registered in 2018 and 2019, respectively,

representing registration rates of 5.74 in 2018 and 5.85 in 2019 per 1000 population. United Nations Population Division projections (based on census and population survey data) of 6.7 and 6.6 Zambian deaths per 1000 in 2018 and 2019[55] suggest these registration rates could constitute 85.7% and 88.6% of total Lusaka deaths respectively. From a median of 302 registrations per week, these data show substantial pre-pandemic volatility (Fig. 3b), exhibiting a range

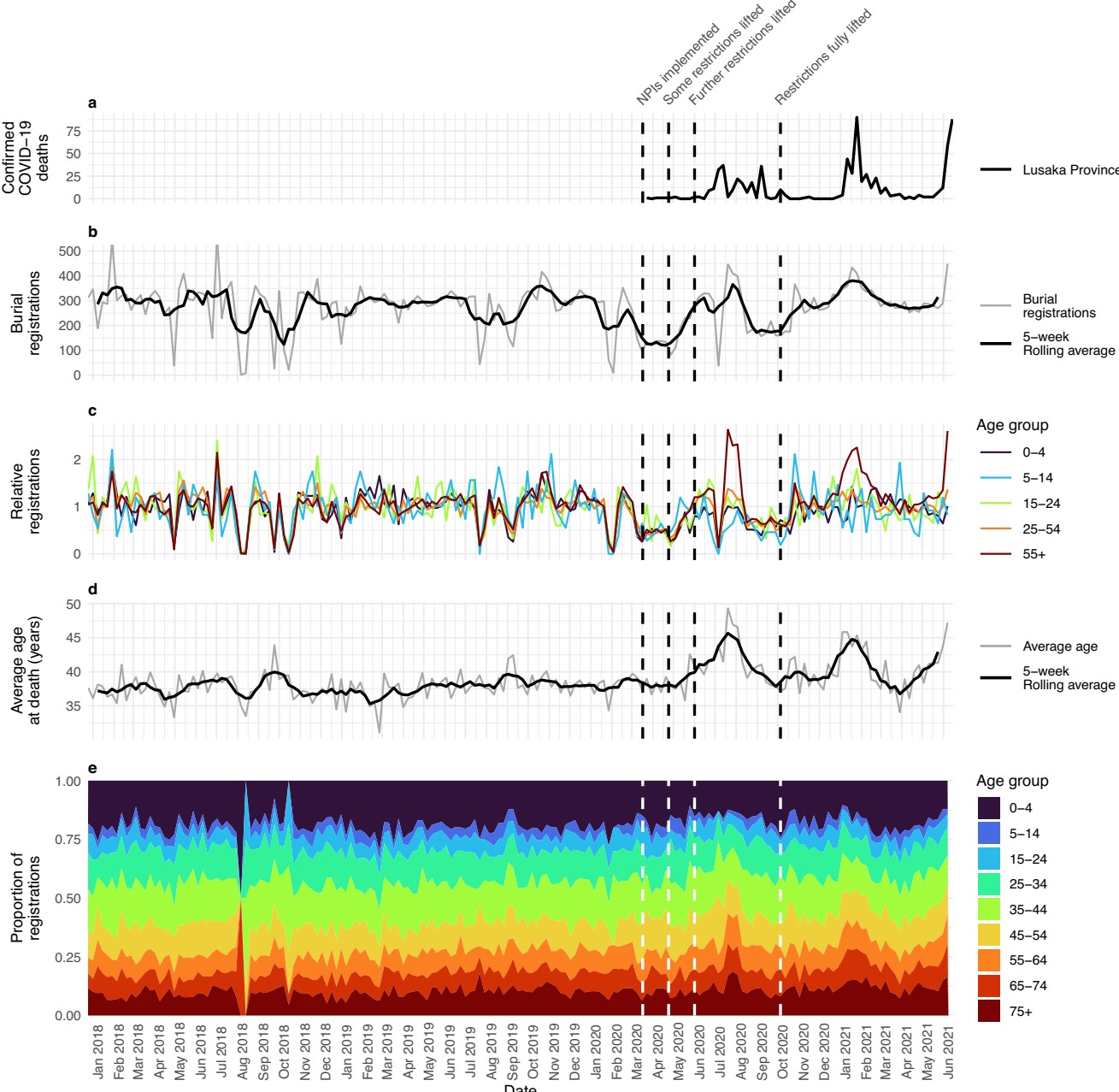

**Fig. 3 | Burial registrations and COVID-19 mortality patterns. a** Confirmed COVID-19 deaths in Lusaka Province, **b** total weekly burial registrations in Lusaka with a 5-week rolling average of the two preceding, current and two succeeding weeks, **c** age-grouped registrations relative to 2018–2019 mean, **d** weekly average age at death of burial registrations with similar 5-week rolling average, **e** age-grouped proportion of deaths in burial registrations. Dates of key

non-pharmaceutical intervention (NPI) changes are also given (vertical dashed lines, 17th March 2020: initial COVID-19 press briefing and NPIs[45], 24th April 2020: initial relaxation of some NPIs[48], 6th June 2020: opening of primary and secondary schools for examination students only[49], 10th October 2020: business restrictions fully lifted, COVID guidance continues, e.g., mask wearing, good hygiene etc[57,85]).

exceeding 500 twice and falling below 100 six times during 2018–2019 weeks. Such variation seems unlikely to represent underlying mortality patterns when non-communicable disease, cancer and HIV/AIDs, accounting for approximately two thirds of Zambian deaths[56], are unlikely to be subject to sharp, temporary declines. When grouped by age and plotted relative to their age-respective 2018-2019 averages (Fig. 3c), these data showed high consistency in pre-pandemic volatility across age-groups. This occurs even as underlying pre-pandemic causes of Zambian mortality vary substantially between age groups. For example, in children under the age of 5 years (U5), maternal and neonatal disorders are the

primary cause of deaths (31.2%) compared to <1% in children aged 5-14; HIV/AIDS is the leading cause of death in children aged 5-14 representing 23.2% of deaths (compared to 13.8% of U5 deaths); injuries represent 14.6% of deaths in 5–14-year-olds but just 2.5% of U5 deaths[56]. In contrast to the weekly registration variability, the pre-pandemic average age at death and age-distribution of burial registrations by date of death are far more stable (Fig. 3d, e), with a median weekly average age of death of 37.6 years (95% CI: 33.9–41.4 years) during 2018–2019. Together, these data suggest that burial registration volatility is caused by service disruption, rather than inherent changes in underlying mortality.

Weekly burial registrations declined across all ages in February 2020, reaching a sustained low during March–April 2020 (Fig. 3b), as awareness of the likely scale of the pandemic intensified and NPIs implemented in Lusaka[44,46]. While the explicit causes of this decline are unknown, trends remain highly consistent across all age groups, with both the burial registration age-distribution and the average age of a registered individual remaining consistent with pre-pandemic levels (Fig. 3c–e). Consequently, and given the aforementioned disparate aetiology of non-COVID-19 mortality across age-groups, we posit that these declines were most likely due to burial registration process disruptions, rather than underlying mortality pattern changes. From late April 2020, as restrictions began lifting, overall registration rates returned to pre-pandemic levels, coinciding with the first recorded epidemic wave. This increase was, however, largely driven by a surge in older-aged registration (Fig. 3c) with the average age at death reaching peaking at 49.3 years old in July 2020, well in excess of the pre-pandemic maximum of 43.9 years. A short, sharp registration decline also occurred during this epidemic wave (in the week of July 6th 2020), though data from this week follow the ongoing temporal trends of increased average age at death (Fig. 3d).

Following the first wave, overall registration rates once again fell below pre-pandemic levels with registration age-patterns, again, consistent with pre-pandemic levels. From late 2020, with the removal of additional restrictions, registrations largely returned to pre-pandemic patterns with the exception of registration surges coinciding with the Beta and Delta COVID-19 waves. Beta-wave registrations peaked at 434 per week in January 2021 and reached 449 per week during the last week of recorded data in June 2021 (mid-Delta wave), approximately 50% higher than the pre-pandemic median. These increases were similarly driven by sharp rises in older-aged registrations, peaking at an average age at death of 45.9 and 51.1 years during these respective waves.

When burial registrations were disaggregated by sex, males were disproportionately represented throughout the time-period (consistent with higher male mortality in Zambia[57]), but essentially no gendered differences are seen in relative registration changes throughout the pandemic compared with the pre-pandemic median (Supplementary Fig. 1).

## Estimating excess mortality in Lusaka during 2020 to mid-2021

The inherent burial registration data volatility limits the utility of basing any predictive model of age-specific mortality trends on the absolute numbers of burial registrations at a given timepoint (Fig. 4a). Instead, we developed a statistical model that attempted to base such predictions on the age-distribution of deaths within those registered for burial. We cross-validated this model, showing that it could generate accurate predictions of the age-distribution of registrations during 2018–2019, stratified by 5-year age groups, (see Supplementary Methods and Supplementary Fig. 2). We then used it to generate predictions, based upon the total weekly U5 burial registrations, of the expected numbers of weekly burial registrations in all other age groups (5+) during January 2020–June 2021 (Fig. 4b). These predictions were compared with the observed data to estimate the excess number of 5+ burial registrations (Fig. 4c). To account for potential differential changes in U5 registration rates (i.e., if neonatal death registrations were differentially affected compared with older age groups), we repeated this process using registration rates from the 5 to 14 year age group to obtain a supplementary set of predictions.

Our approach showed that total and age-specific registration declines during the implementation of NPIs followed a predictable pattern based upon U5 registration declines in the same time-period. However, a clear pattern of higher-than-expected 5+ registration became apparent following the lifting of restrictions. Increased excess registration subsequently followed the three known waves of the pandemic contemporary with our data, with a non-linear per-capita increase with age (Fig. 4c). This approach produced an estimated 2332 (95% Credible Interval (CrI): 1719–2924) excess 5+ burial registrations during January 2020-June 2021, of which 1651 (95% CrI: 1209–2078) occurred during 2020. These estimates correspond with 644.1 (95% CrI: 471.6–810.7) and 909.7 (95% CrI: 670.6–1140.7) excess registrations per million total population during 2020 and during January 2020–June 2021 respectively. The estimates also represent 10.3% (95% CrI: 7.6–12.9%) and 10.5% (95% CrI: 7.7–13.3%) of median pre-pandemic burial registrations burial registrations during 2020–June 2021 and 2020, respectively (Table 1). Our supplementary analysis, using registration trends in 5–14 year olds to estimate expected trends in older ages, produced similar results (Supplementary Figure 3).

We then applied a weekly scaling factor, equivalent to the relative difference between reported U5 weekly mortality and the median U5 registration rate during 2018–2019, to these estimates to account for our posited most plausible assumption (i.e., temporary changes in burial registration are primarily driven by registration process factors, rather than underlying changes in non-COVID-19 deaths) (Fig. 4d, e). This scaling produced cumulative estimated excess deaths of 2898 (95% CrI: 2031–3953) during 2020 and 3977 (95% CrI: 2931–5205) during 2020–June 2021 within the pre-pandemic proportion of the Lusakan population whose deaths would typically be registered (Fig. 4f).

These numbers therefore correspond to the most optimistic registration system assumptions (i.e., a 100% pre-pandemic registration rate), yielding 1551.5 (95% CrI: 1090.0–2097.2) deaths per million Lusakans throughout the study period. Though accurately quantifying this system coverage, herein referred to as 'capture rate', is impossible with available data, we make a baseline assumption, using local knowledge and census-based total mortality projections in Zambia, that approximately 90% of pre-pandemic deaths were registered (Fig. 4e, f). With this assumption, we estimate a total excess mortality of 3220 (95% CrI: 2256–4393) during 2020 and 4419 (95% CrI: 3257–5783) during 2020–June 2021, corresponding to 1256.2 (95% CrI: 880.1–1713.8) and 1723.9 (95% CrI: 1270.6–2256.0) deaths per million total population, respectively. A more pessimistic assumption of burial registration population coverage yields correspondingly more pessimistic excess mortality estimates (e.g., assuming 80% coverage produces an estimated 4971 (95% CrI: 3664–6506) excess deaths throughout the study period, Table 1). Our excess death estimates represented 18.5% (95% CrI: 13.0–25.2%) and 17.6% (95% CrI: 13.0–23.0%) of pre-pandemic burial registrations for the two respective time-periods, exceeding 50% of 2018–2019 median registrations (Supplementary Fig. 4, approaching 150% when filtered to deaths over 50 years) during the peaks of all three waves, are robust to this registration coverage uncertainty. Finally, we used these estimates to calculate DVWI values, finding that all these values exceeded comparative values based on WHO excess mortality estimates, with median excess mortality estimates (i.e., using the scaling factor) over twice as large as comparative WHO median for any capture rate (Fig. 4g).

## SARS-CoV-2 transmission and COVID-19 severity during the first wave

We developed a Bayesian inferential framework that allowed us to fit an existing model of SARS-CoV-2 transmission dynamics[58], parameterised to Zambia's demographic structure, to our estimates of mortality patterns and available data on infection prevalence within Lusaka during the first pandemic wave. The model was fit to burial registry data by combining our non-COVID-19 baseline registration estimates with modelled COVID-19 deaths, scaling those modelled deaths using weekly U5 registrations relative to their 2018–2019 median, and using a default assumption of 90% capture of deaths within the Lusaka registration system. The model was also fit to post-mortem PCR prevalence by age and week[27], accounting for likely

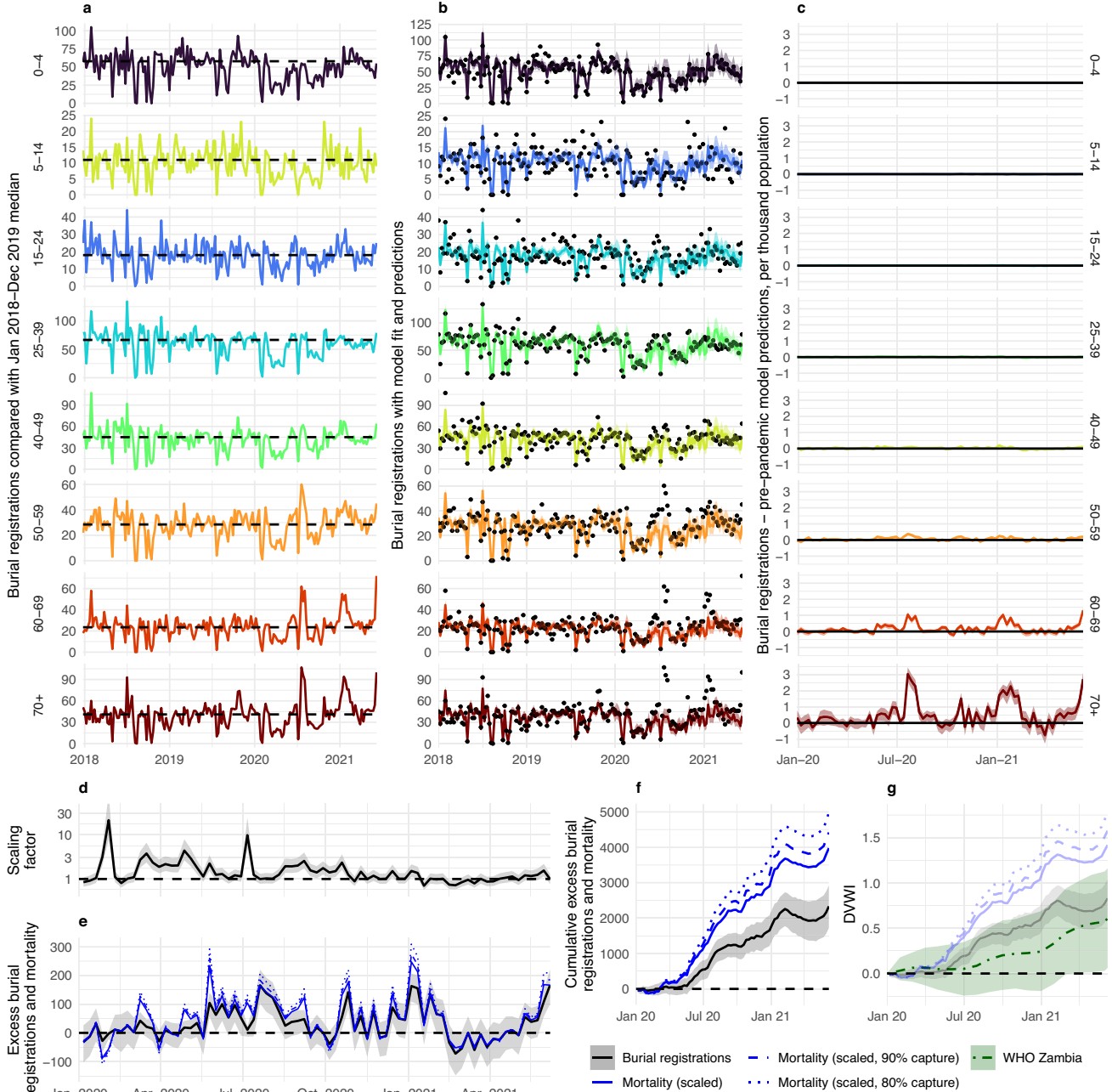

**Fig. 4 | Excess mortality in Lusaka.** Figure shows **a** burial registrations grouped by age with 2018–2019 median shown throughout (dotted line), **b** burial registrations (points) with model fit (line and ribbon) combining age-distribution of deaths with the number of registrations aged <5 fitted to 2018–2019 burial registrations and predictions of expected registration in 2020–2021, **c** excess burial registrations per thousand people based on the difference between total burial registrations and 2020-21 model predictions, **d** scaling factor based on burial registrations aged <5 relative to their pre-pandemic median, **e** application of scaling factor to excess burial registration of population aged 5+ (black) to estimate median excess mortality (blue) and with additional assumption of 90% and 80% registration capture of underlying mortality (blue, dashed and dotted lines), **f** cumulative estimates of

excess burial registrations (black), median cumulative mortality assuming weekly scaling (blue), and median cumulative mortality assuming 90% and 80% registration capture of underlying mortality (blue, dashed and dotted), **g** demographic-vulnerability-weighted index (DVWI, the cumulative attack rate required to achieve excess burial registrations (black), or mortality assuming scaling (blue) with 100%, 90%, and 80% registration capture of underlying mortality registration (solid, dashed and dotted) in **e**, **f** or in World Health Organisation (WHO) excess mortality estimates for Zambia, assuming the overall IFR for Lusakan and Zambian population structures, respectively, and direct COVID-19 causation. In all model plots, lines and ribbons show the median and 95% credible interval. Registrations, mortality and DVWI are grouped by week in all panels.

patterns of PCR positivity in individuals whose deaths were not caused by COVID-19, and to population-based PCR prevalence and seroprevalence survey data[37]. We fitted a time-varying reproduction number ($R_0(t)$) at two-week intervals, as well as the start date of the epidemic. Using our current globally-derived estimates of age-specific IFR[42], we obtained a qualitatively good fit to the data, with the model

largely able to replicate burial registration patterns while maintaining infection levels consistent with population survey and mortuary-based post-mortem sample data (Fig. 5a–e). The relatively flat population-based spread by age within the model is also, to the extent it can be given the small sample sizes due to age-stratification, supported by the data (Supplementary Fig. 5).

**Table 1 | Excess burial registrations and mortality estimates in Lusaka**

| | 2020 | | 2020–June 2021 | |
|---|---|---|---|---|
| | Median | 95% CrI | Median | 95% CrI |
| **Excess** | | | | |
| Burial Registrations[a] | 1651 | 1209–2078 | 2332 | 1719–2924 |
| Mortality[b]: 100% capture[c] | 2898 | 2031–3953 | 3,977 | 2931–5205 |
| *Mortality: 90% capture* | **3220** | **2256–4393** | **4419** | **3257–5783** |
| Mortality: 80% capture | 3622 | 2538–4942 | 4971 | 3664–6506 |
| **Excess per million** | | | | |
| Burial Registrations | 644.1 | 471.6–810.7 | 909.7 | 670.6–1140.7 |
| Mortality: 100% capture | 1130.5 | 792.3–1542.1 | 1551.5 | 1143.4–2030.5 |
| *Mortality: 90% capture* | **1256.2** | **880.1–1713.8** | **1723.9** | **1270.6–2256.0** |
| Mortality: 80% capture | 1413.0 | 990.1–1713.8 | 1939.3 | 1429.4–2538.1 |
| **% of increase in registrations/deaths relative to pre-pandemic levels** | | | | |
| Burial Registrations | 10.5 | 7.7–13.3 | 10.3 | 7.6–12.9 |
| *Mortality (irrespective of capture)* | **18.5** | **13.0–25.2** | **17.6** | **13.0–23.0** |
| **DVWI** | | | | |
| Burial Registrations | 0.589 | 0.431–0.742 | 0.832 | 0.613–1.043 |
| Mortality: 100% capture | 1.034 | 0.725–1.411 | 1.419 | 1.046–1.857 |
| *Mortality: 90% capture* | **1.149** | **0.805–1.568** | **1.577** | **1.162–2.064** |
| Mortality: 80% capture | 1.293 | 0.906–1.763 | 1.774 | 1.308–2.322 |

[a]'Burial registration' refers to our estimates of excess registration in older ages (5+ years), calculated relative to expected baseline registrations derived from weekly registration patterns in the <5 years age group, assuming that these baseline estimates reflect underlying non-COVID-19 deaths across all ages. [b]'Mortality' refers to our estimates of excess mortality assuming changes in <5 years mortality are driven by changes in the likelihood that deaths are reported rather than underlying changes in non-COVID-19 deaths. [c]'Capture' refers to the proportion of deaths in Lusaka typically captured within the system prior to the pandemic. *refers to the percentage of all registrations/deaths within a given time-period which were deemed excess. Highlighted in bold and/or italics are results using our default assumptions that 90% of underlying mortality is captured by pre-pandemic burial registration.

Transmissibility through time is given in Fig. 5f, with modelled estimates of time-varying basic reproduction number ($R_t$) analogues: $R_0(t)$ and $R_{eff}$ (shown at 50% and 95% credible intervals). $R_0(t)$ is a measure of secondary infections that would occur in a wholly susceptible population, only capturing contact rate changes throughout the epidemic. $R_{eff}$ additionally incorporates population immunity effects on transmission such that $R_{eff} > 1$ indicates a growing epidemic. These results show that $R_0(t)$ and $R_{eff}$ trends begin at around three before the first epidemic wave, falling to below one in mid-July 2020 (Fig. 5f), consistent with the gradual post-mortem prevalence decrease through time. $R_0(t)$ and $R_{eff}$ remain close to one another for the remainder of the study period, suggesting a limited role of population-level immunity role in reducing transmission, with an estimated 24% attack rate by October 2020 (Fig. 5c).

We then assessed the model goodness of fit under differing age-gradient and overall COVID-19 IFR assumptions during Lusaka's first epidemic wave (Fig. 6a, b). The log average posterior model fit to these data proved more sensitive to the IFR age-gradient than the overall Lusaka population-level IFR assumed, but centred largely upon estimates similar to our default model assumptions (Fig. 6c–e), with anything beyond a halving or doubling of either parameter producing quantitatively and qualitatively worse fits to the data (Supplementary Fig. 6). Where Zambia's demography-weighted IFR under default assumptions is 0.11%, this range (80–167% of default overall IFR assumptions) corresponds with an overall population IFR between 0.088 and 0.183%.

We tested our assumptions through several sensitivity analyses (Supplementary Table 1, Supplementary Fig. 7). These included verifying that our results were not sensitive to the unexpectedly high post-mortem prevalence during 13th–19th July, 2020, compared with preceding and succeeding weekly prevalence (Supplementary Fig. 7b). Our default fits, which assume near-zero COVID-19 U5 mortality, produced lower-than-observed U5 mortuary prevalence; a sensitivity analysis excluding U5 mortuary prevalence demonstrated fits

favouring marginally higher IFRs in the remaining 5+ population (Supplementary Fig. 7c). To assess the sensitivity of our results to our assumptions around declines in overall burial registration rates, we also refitted our model to our excess mortality estimates without applying the scaling factors used in our default approach and censoring data during acute burial registration service disruption during 6th–19th July, 2020. We found this approach favoured marginally higher IFRs than our default set of assumptions (Supplementary Fig. 7d).

We found limited sensitivity of our results to duration between infection and death, though an increase in duration again favoured marginally higher IFRs (Supplementary Fig. 7e, f). Furthermore, we conducted a sensitivity analysis of our underlying assumption that the pre-pandemic relative rates of 5+ baseline non-COVID-19 mortality remained consistent with U5 mortality during the pandemic. We did this by varying the rates of estimated pre-pandemic non-COVID-19-related mortality in 5+ relative to U5s by ±10% (e.g., either a differential reduction in U5 mortality greater than would occur if all deaths due to respiratory illness in U5 fell to zero or a differential reduction in 5+ mortality greater than all deaths from injuries in 5+ falling to zero[56]) and by ±20% (e.g., either a differential reduction in U5 mortality greater than would occur if all deaths due to respiratory and diarrhoeal illness or a differential reduction in 5+ mortality greater than deaths from injuries and HIV falling to zero[56]). We found that a 10% rate increase or decrease produced only nuanced impact upon the distribution of the fit to different IFR patterns, with our default parameters remaining well within the envelope of best-fitting values (Supplementary Fig. 7g, h). Meanwhile, a 20% decrease in relative non-COVID-19 5+ mortality produced substantially higher IFR estimates, excluding our default IFR assumptions and including a fit with flatter IFR age-gradient (Supplementary Fig. 7i). A 20% increase produced lower estimates (although still solidly encompassing our default IFR patterns) and included produced best-fitting scenario with a steeper IFR age-gradient (Supplementary Fig. 7j). Finally, we varied our 90%

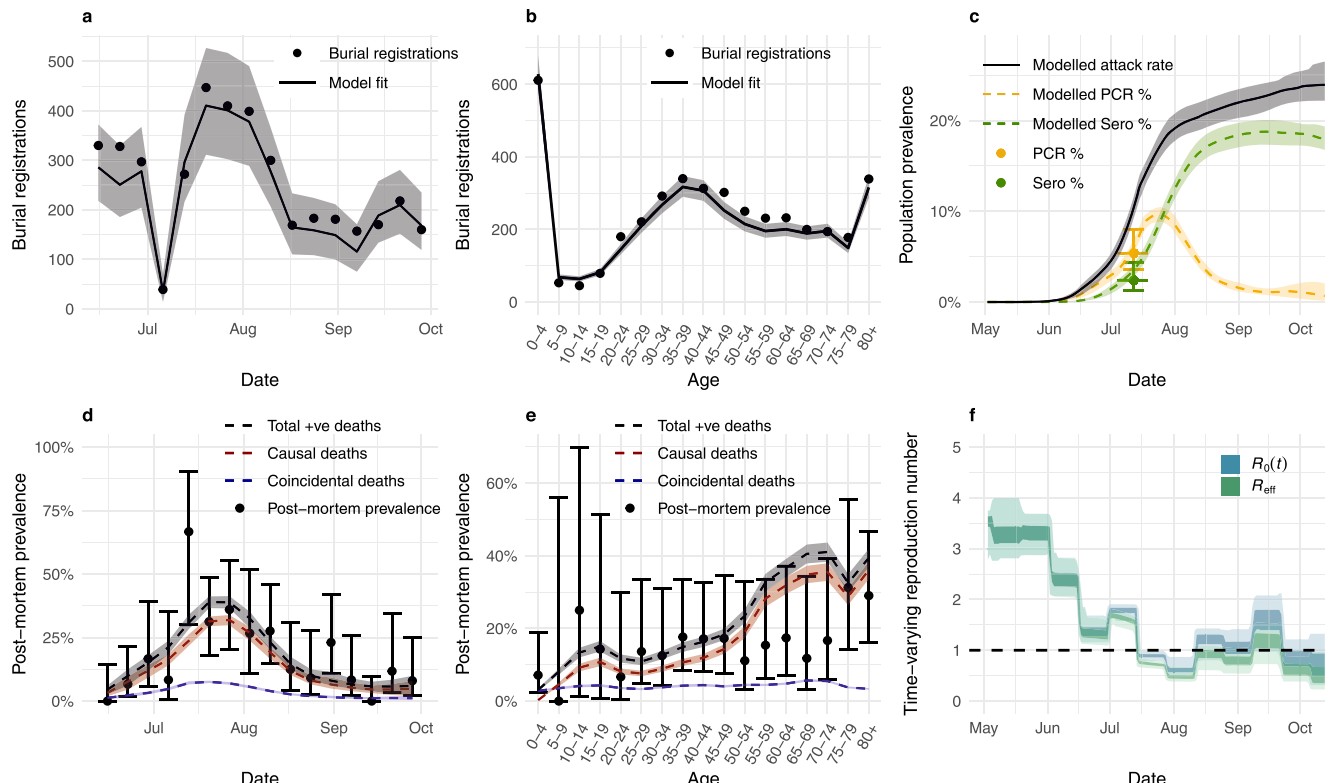

**Fig. 5 | SARS-CoV-2 transmission in June–October 2020.** Transmission model fit to burial registrations by **a** week and **b** 5-year age group during 15th June–4th October 2020, **c** Lusaka population-level polymerase chain reaction (PCR) prevalence and seroprevalence (sero) surveys, and post-mortem PCR prevalence at UTH with 95% binomial confidence intervals by **d** week and **e** 5-year age group during 15th June–4th October 2020, showing total positive (+ve) deaths, delineated by causal and non-causal COVID-19 deaths. Lines and ribbons in panels (**a**–**e**) show median and 95% credible intervals of 100 samples. Transmissibility through time is given in **f** with the time-varying reproduction number $R_0(t)$, and effective reproduction number $R_{eff}$ as a comparable measure incorporating the impact of population immunity (shown at 50% and 95% credible intervals of 100 samples) such that a value greater than one indicates a growing epidemic.

burial registration capture rate assumption, again finding nuanced differences, with a small increase in severity with 80% burial registration capture (Supplementary Fig. 7k–l).

Within the best-fitting-model envelope (Supplementary Fig. 8), infection spread and transmissibility estimates are similar to those of the default model (Fig. 4), suggesting a reproduction number well above two during May-June 2020 but falling progressively to below one by late July 2020 as the epidemic peaked. Our estimates suggest that this transmission decline occurred despite population-level immunity remaining low, with our cumulative attack rate estimates (the % of the population infected at any point during the wave) ranging between 15 and 30% during the first wave.

## Discussion

Uncertainty in mortality patterns in Africa throughout the pandemic is one of the leading contributors to the remaining uncertainty in the impact of the pandemic globally[33,59–61]. While the findings presented here are not readily extrapolated to a wider region due to substantial heterogeneity between countries[33], our COVID-19 impact estimates in Lusaka represent a substantial narrowing of uncertainty in one of the many countries in Africa where current estimates of impact are largely uninformative and where age demographics are amongst the most favourable globally in terms of reducing the average likelihood of severe disease upon infection.

Our results strongly suggest that the first COVID-19 wave in Lusaka had a direct and heavy impact, shifting the age-distribution of mortality towards older ages in a manner highly characteristic of COVID-19 severity patterns observed elsewhere. Assuming U5 burial registration declines represent registration process rather than underlying

mortality changes, our estimates of 3220 (95% CrI: 2256–4393) excess deaths in 2020 represent a per-capita rate of 1256.2 (95% CrI: 880.1–1713.8) excess deaths per million, approaching the highest affected Africa-region countries (South Africa: 920 (95% CI: 830–1000) per million, and Algeria: 1270 (95% CI: 1210–1340) per million). After accounting for the protective effect of Lusaka's young population, these estimates far exceeded those measured for Europe and the Americas (DVWI: 1.149 (95% CrI: 0.805–1.568) compared with 0.171 (95% CI: 0.167–0.176) and 0.240 (95% CI: 0.233–0.248), respectively). Assuming, instead, that registration declines are driven by underlying mortality changes decreased our excess 2020 mortality estimates by 48.7%, although estimates accounting for Lusaka's population demographic remained well above those in the aforementioned countries (DVWI: 0.589 (95% CrI: 0.431–0.742)). In contrast, our default estimates may be conservative if, for example, disruptions to burial registrations were mirrored by, and subsequently mask any impact thereof, any disruption to maternal and neonatal services[62]. U5 registration patterns, however, correlated well with older children and younger adults (despite having different typical mortality causes) and were highly predictive of trends across all age groups when NPIs were implemented, with similar results using the 5–14 year age group as a reference category, suggesting that mortality changes due to behavioural factors may have been relatively nuanced in the short term.

Using a transmission model parameterised by IFR patterns estimated from available global 2020 data[42], we show that these mortality patterns correspond well to community and post-mortem mortuary prevalence data, with these IFR patterns well within the range of our best-fitting models. Consequently, we find no evidence that age-specific severity was markedly different from estimates in other

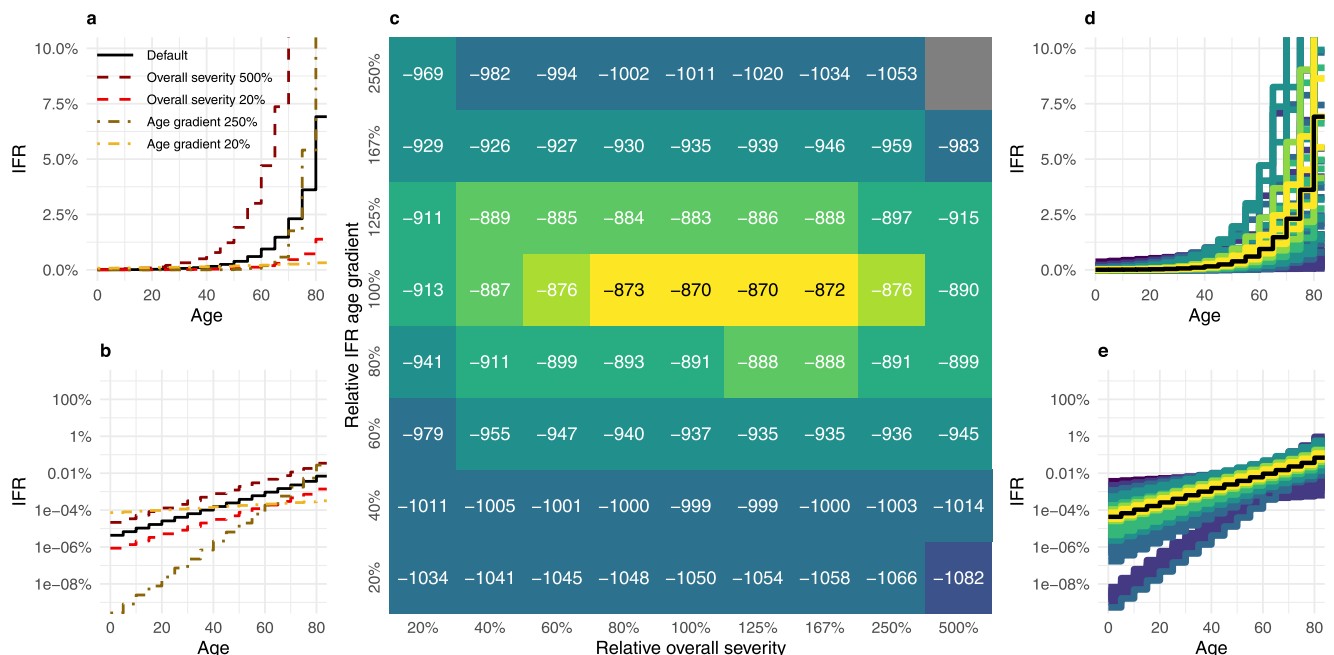

**Fig. 6 | Inference of age-gradient and scale of severity. a**, **b** are infographics to show how the infection fatality ratio (IFR) curve changes when the intercept or slope is altered on **a** standard and **b** log scales. Each plot shows the default IFR from Brazeau et al.[42] as a solid line, with relative overall severities of 20% and 500% of those default values or relative age-gradient of 20% and 250% of the slope on the log scale, maintaining the overall severity of the default. The heatmap **c** shows the log of the average posterior model fit over 100 samples. **d**, **e** show all assessed IFR curves, coloured by posterior fit as found in **c**, and where default IFR assumptions are highlighted in black, plotted on **d** standard and **e** log scales.

geographies, or any support for a so-called "Africa paradox"[12,16,17]. Our results do not, however, preclude relatively nuanced IFR differences in Lusaka relative to those observed elsewhere, with our results showing a plausible relative severity range between 50 and 250% of our default assumptions across numerous sensitivity analyses. Indeed, given the high non-hospital death prevalence[27], we might expect to see greater severity in Lusaka compared with estimates from settings with high hospital access under the same intrinsic disease pathogenicity assumptions. IFR patterns toward our uncertainty interval's higher end would also bring our estimates more in line with others from low- and middle-income settings outside of Africa[29,36]. Alternatively, IFR patterns toward the interval's lower end could suggest relatively nuanced intrinsic severity differences in Lusaka that remain unexplained. However, given the widespread low hospital-care access across large parts of Africa, such an observation would not support a low direct pandemic impact across the continent. Meanwhile, COVID-19 treatment advances throughout 2020[63], largely benefitting high-income-country patients, are likely to mean that, even at the most optimistic end of our uncertainty, prioritising prevention efforts in higher-income-setting individuals over equivalently-aged Lusakans could have no equitable justification.

Our default IFR assumptions are based on results from Brazeau et al.[42], calculated using data matching our main study period time-frame (i.e., prior to the emergence of new variants of possible differing severity). As with all severity studies during the initial pandemic stages, this study included data representing a trade-off between study-design quality and the representation of a wide range of contexts, leading to data inclusion with some potential measurement error. Bias may therefore be present in some population exposure data (i.e., data collected through convenience sampling including shopping centre attendees and blood donors) and COVID-19 mortality data (i.e., where confirmed COVID-19 mortality use can underestimate total attributable mortality). However, the included data come from countries with strong testing systems, and crude IFR estimates from convenience-sample sources are not dissimilar to other included estimates. Other

studies have suggested a higher IFR for very young children relative to older children[64–66], which might account for the high observed U5 post-mortem prevalence, though at levels (<0.01%) that would make negligible difference to the fit of our model to the data. A plausible explanation for this U5 prevalence, though not one quantifiable in our framework, could be comparatively extensive SARS-CoV-2 spread within communities of high non-COVID-19-associated infant mortality. Overall, Brazeau et al. IFR estimates are central within the range of other estimates[42]. Thus, it seems plausible that an ensemble approach could broaden our uncertainty but would be unlikely to alter our central conclusion that, when analysing one of the best-characterised epidemics in sub-Saharan Africa, there is no evidence to support any substantial differences between innate COVID-19 severity in Lusaka relative to estimates from other parts of the world.

When considering SARS-CoV-2 spread, our estimates suggest that the potential transmissibility in Lusaka was comparable to estimates from earlier epidemics in China, Europe and America[67–71] but that initial dynamics were comparable to those in South Africa[72], with NPIs implemented at a very nascent stage in the epidemic, delaying the emergence of a first observable wave until restriction relaxation began in mid-2020. Following this initial exponential growth period, our estimates suggest that control of the epidemic was re-established around August 2020, showing patterns that closely resemble trends in Malawian prevalence data[73]. As a result, whilst our estimates suggest substantial direct pandemic impact, they also suggest that greater potential impact was mitigated by measures taken by the Lusakan population throughout 2020, despite the relaxation of official restrictions[48–51].

Unfortunately, our estimates suggest that by the year's end most of the population remained entirely immune-naïve, suggesting that Lusakans remained highly vulnerable to future waves, even in the absence of the variants that subsequently emerged. Our estimates of subsequent peaks in excess mortality suggest that many gains achieved by control measures during 2020, similarly implemented in many African countries[74], were lost later in 2020 and 2021. Given vaccine availability delays in Zambia throughout 2021[75,76], maintaining

these early gains would have involved maintaining costly suppression measures well beyond the duration they needed to be maintained in higher-income countries where vaccine distribution was prioritised. The unsubstantiated perception of a low COVID-19 severity across Africa seems unlikely to have helped advocacy for equitable vaccine access, nor combat vaccine hesitancy in many African countries when vaccines were introduced[77,78]. In this context, a global inequity in our ability to measure mortality patterns and disease spread in 2020 likely contributed substantially to global inequity in the impact of the pandemic.

## Methods

### WHO estimates of excess mortality in Zambia in 2020

Estimates of excess mortality during 2020 were sourced from the WHO[33] and plotted per million population with 95% confidence intervals at global and WHO-region level, and at national level within the Africa WHO region. Confirmed WHO COVID-19 mortality data for countries and regions are plotted alongside for ref. 2.

Population-weighted country and region specific overall IFR were calculated and plotted using 100,000 samples from the age-specific IFR distribution, derived by Brazeau et al.[42] (without seroreversion), which we weighted by the specified population structure and summed to generate an overall value with 95% credible intervals. Population demographic estimates were sourced from the UN World Population Prospects[79] and grouped into 5-year age brackets, then summarised at the global, regional and country level (consistent with excess mortality estimates).

DVWI was defined as the cumulative population attack rate required to directly generate the estimated excess mortality impact, given the median global, region or country-specific overall IFR and assuming an even spread of infection by age (for negative values of excess mortality this DVWI was set to 0). We calculated these DVWI values by dividing estimates of excess mortality by median overall area-specific IFR estimates to generate a population infection level reflective of area demography and mortality.

### Model framework and fitting

**Estimating excess mortality in Lusaka using burial registration data.** Official reported deaths for Lusaka Province were obtained from the Zambia COVID-19 Dashboard[43], while details of NPIs were obtained from situation reports from the Zambia National Public Health Institute website[80] and governmental statements on the COVID-19 pandemic[44–51]. Calculations involving Lusaka population size and demography were obtained from the Zambia Statistics Agency[81], which are recently updated census-based projections.

Excess mortality in 2020–2021 was estimated by comparing age-stratified all-cause weekly burial registrations, which dataset begins in mid-2017 and ends in mid-2021, sourced from the UTH burial permit office[39] with predictions of baseline registration based on pre-pandemic registrations (2018–2019). Although the burial registration dataset begins in 2017, we found that registrations in 2017 increase from low numbers, only reaching greater stability and reliability from 2018, and therefore censored registrations during 2017 in our analysis. Based upon local knowledge of the study team, estimates from the literature[53,54] and a comparison of recorded burials in 2019 with census-based projections[81] we made a default assumption of a 90% burial registration rate (with sensitivity analyses of 80–100% registration rate around this estimation in our transmission modelling). We used a Bayesian framework and Metropolis-Hastings Markov chain Monte Carlo (MCMC) to model baseline registrations as a function of weekly 2020–2021 registrations in the youngest age group and age-specific relative rates of registration trained on 2018–2019 data, using the *drjacoby* R package[82]. We assume that non-COVID-19 related registrations in each age-week group follow a Poisson distribution, verifying no better fit could be obtained using a Negative Binomial distribution with

various assumed levels of over-dispersion (Supplementary Fig. 9), with mean equal to the product of weekly U5 registration, with a uniform prior, and the age-specific rate for each age group, relative to the U5 reference category and otherwise given a diffuse log-normal prior centred around one in order to give equal prior weight to an increase or decrease. We also estimated parameters for U5 registration for each week in 2020–2021, again assuming a Poisson distribution with a uniform prior on registrations. Following a burn-in of 5000, we drew 3000 samples from each of five chains for each parameter. By assuming that COVID-19 had little impact on U5 mortality and that age-specific rates of deaths are constant through changes in total mortality (see Supplementary Methods, Supplementary Fig. 2), we multiplied these relative age-specific rates by weekly U5 registration rates in 2020–2021 to obtain baseline predictions of registration. To account for deficiencies in burial registration data, we scaled estimated excess registrations by the ratio of total U5 registrations in that week to the median weekly U5 registrations from 2018 to 2019 and assuming that 90% of deaths in Lusaka are registered. We also tested our dependence on the U5 age group as a baseline by conducting the same analysis using the 5–14 years age group (Supplementary Fig. 3).

**Modelling SARS-CoV-2 transmission.** SARS-CoV-2 transmission in Lusaka's first wave of the pandemic (June-October 2020) was modelled using an age-structured SARS-CoV-2 SEIR model[58] (Supplementary Figure 10) with age-specific population estimates for Lusaka District obtained from the Zambia Statistics Agency[81]. In the absence of locally collected data on social contact patterns we used a social contact matrix generated from data collected in Manicaland, Zimbabwe, the nearest geographical location in the literature with a matrix that describes contacts across all ages of interest, filtering to only include data from the peri-urban region (Nyanga) within the dataset[83]. As is generally the case in data collected from lower-income countries[84], this matrix produces attack rates throughout an epidemic which are much flatter by age then any equivalent simulation using data from higher-income settings. For validation, PCR prevalence and seroprevalence patterns by age as observed in the population-based survey in Lusaka were compared with those estimated contemporaneously by the model. We also adjusted standard parameterisation to account for some uncertainties in treatment access in Lusaka (see Supplementary Methods, Supplementary Table 2).

The model was fitted to age-specific weekly post-mortem PCR prevalence, collected at UTH during the first pandemic wave (June 15th-October 4th, 2020) and shared by the COVID-19 extension of the ZPRIME study[27], age-specific weekly burial registrations (described above, limited to the same dates as the post-mortem PCR prevalence study, June 15th-October 4th 2020) and Lusaka-specific population level PCR prevalence and seroprevalence surveys in July, 2020, shared by the Centers for Disease Control and Prevention, Lusaka, Zambia[37].

We again assume that all-cause burial registrations follow a Poisson distribution (with lambda equal to the sum of model predicted COVID-19 deaths, scaled to burial registry level as described above, and estimated baseline burial registrations, integrated over 4000 baseline estimation samples). We also assume that post-mortem PCR prevalence follows a binomial distribution (with probability equal to the modelled prevalence of PCR positive deaths and samples equal to the number of tests conducted) and that positive population-level PCR and serological tests both follow Binomial distributions (with probabilities equal to the average respective modelled PCR prevalence and seroprevalence over the data collection time, incorporating changes in test detection probability as function of time since infection, Supplementary Figure 11). We used a Bayesian framework with Metropolis-Hastings MCMC based sampling scheme, drawing 30,000 parameter sets from each of eight MCMC chains that were used to simulate the epidemic wave (with the first 10,000 discarded as burn-in, see Supplementary Table 3 for priors used).

**Inferring COVID-19 severity in Lusaka.** Severity of COVID-19 was inferred by adjusting the default IFR values. Default IFR inputs are taken from Brazeau et al.[42] and follow a log-linear relationship. We varied the log-linear curve by intercept (to increase and decrease overall IFR) and by slope (to change the age-gradient of IFRs while maintaining the same overall IFR value). For each intercept-slope combination, we ran our model fitting as described above, comparing the mean likelihood and posterior values of 100 draws from eight chains, each of 30,000 MCMC samples (with the first 10,000 discarded), to identify the combination that allowed the best fit to the data. A range of sensitivity analyses (see Supplementary Table 1) were then performed to assess the influence of key outlying datapoints, our assumptions around the delay distributions between infection and deaths and our assumptions around the translation of burial registration data to excess deaths, namely: i) removing our scaling factor (i.e., assuming that changes in U5 registrations reflect changes in underlying non-covid mortality rather than disruption to registration processes); ii) assuming different levels of shifts (either ±10% or ±20%) in the ratio between U5 and 5+ non-COVID-related mortality relative to that observed prior to the pandemic; iii) varying the pre-pandemic registration rate between assuming 80% and 100% of all deaths in Lusaka.

Additional details of methods are given in Supplementary Information.

## Ethics and inclusion statement

The collaboration underpinning this analysis began after PGTW was approached by a journalist looking to understand the impact of Covid-19 in Africa who then suggested that they speak to LM and CJG whose team had recently published a study looking at the prevalence of the virus within the main morgue of Lusaka. Meetings between PGTW, LM, CJG and OJW then began after LM expressed an interested in understanding whether dynamical modelling could help to provide insight into the observed patterns within the mortuary data. Subsequently, the analysis plan was developed through regular calls and updates between investigators with the implementation of several skill-sharing activities. This included RJS travelling to Lusaka to meet the study team and to engage in collaboration with researchers from Zambia National Public Health Institute (ZNPHI) and to contribute to a two-week modelling course designed to help support Zambian-based researchers to conduct their own modelling analyses. Most recently, SLC has visited the Department of Infectious Disease Epidemiology at Imperial College to share insights into the impact of COVID-19 more broadly across Zambia and to promote further collaboration in the form of twice-monthly meetings between study teams on an ongoing basis.

We are not aware of any restrictions or prohibitions that have applied to the work of local researchers in this analysis.

Ethical oversight for ZPRIME and the COVID-19 expansion that generated post-mortem PCR prevalence data from UTH[27] were provided by the institutional review boards at Boston RESEARCH University and the University of Zambia. Written informed consent was obtained from the deceased's family members or representatives.

The population level SARS-CoV-2 prevalence study[37] was approved by the Zambia National Health Research Authority and the University of Zambia Biomedical Research Ethics Committee. The study was reviewed in accordance with the Centers for Disease Control and Prevention (CDC) human research protection procedures and was determined to be non-research. Written informed consent was obtained for adults (aged ≥18 years) and emancipated minors, parental consent was obtained for participants aged 17 years and younger, and assent was obtained for participants aged 7–17 years, before the study.

The work was granted approval via Imperial's Research Governance Integrity framework on the basis of the above pre-existing ethics approvals.

We are not aware of any personal risks to participants, but we are eternally grateful to them. In particular, we are grateful to the families who consented to the collection of post-mortem samples from loved ones during some of the hardest times imaginable and to these data being used within secondary analyses such as ours.

This analysis prominently cites and builds directly upon, world-leading locally-generated research, already published in leading peer-reviewed journals and would not have been possible without it.

## Reporting summary

Further information on research design is available in the Nature Portfolio Reporting Summary linked to this article.

## Data availability

The WHO excess mortality data and UN World Population Prospects demographic data that support the findings of this study are available from https://www.who.int/data/sets/global-excess-deaths-associated-with-covid-19-modelled-estimates and https://population.un.org/Dataportal/ (accessed 11th May 2022), respectively, and may be found in the project repository along with the IFR estimates used to generate Fig. 2, the Nyanga contact matrix and aggregated (by week and age group) versions of the burial registration, post-mortem prevalence, and population survey datasets. Full burial registration and post-mortem prevalence data may be shared in full, on the basis of a request through a formal data sharing agreement between relevant authors. Deidentified population survey participant data used for this analysis can be requested from the Zambian Ministry of Health, along with the unpublished demography estimates for Lusaka used in this study (interested researchers must submit a research proposal for consideration by the original study investigators. If approved, the requestor must sign a data use agreement). All data requests should be directed to the corresponding author who will facilitate the initiation of relevant processes within two weeks.

## Code availability

All the included data and code with instructions for the reproduction of these results can be found at https://github.com/RJSheppard/COVID.IFR.Lusaka (https://doi.org/10.5281/zenodo.7963552).

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

## Acknowledgements

This work was supported by funding from the MRC Centre for Global Infectious Disease Analysis (reference MR/R015600/1: R.S., O.J.W., G.B., I.C.G.G., D.O.M., C.W., S.G., L.C.O., A.C.G., P.G.T.W.), jointly funded by the UK Medical Research Council (MRC) and the UK Foreign, Commonwealth & Development Office (FCDO), under the MRC/FCDO Concordat agreement, and is also part of the EDCTP2 programme supported by the European Union and funding by Community Jameel. The research has also been supported in part by the President's Emergency Plan for AIDS Relief (PEPFAR) through the Centers for Disease Control and Prevention (CDC) through a cooperative agreement with the Zambia Ministry of Health: buffer component COVID-19 response funding No: CDC-FRA-GH15-160005CONT19, "Strengthening the Zambian Ministry of Health's capacity to provide Leadership to the National COVID-19 Response through Coordination, Policy Development, Mentorship and Training under the International Task Force" (Cooperative Agreement number: 5 NU2GGH001617-05): J.Z.H., L.B.M. The ZPRIME study, COVID-19 expansion and further support of these analyses were funded by the Bill & Melinda Gates Foundation (OPP 1163027: R.P., J.L., G.K., C.M., W.B.M., L.M., C.J.G.). Additional funders included The Wellcome Trust and the UK Foreign, Commonwealth & Development Office (FCDO) (reference 221350/Z/20/Z: P.G.W., O.J.W.), an Academy of Medical Sciences Springboard Fellowship (reference SBF005\1107: P.G.W., R.S.), a Schmidt Science Fellowship in partnership with the Rhodes Trust and the Centers for Disease Control and Prevention of the U.S. Department of Health and Human Services (HHS) as part of financial assistance award (reference U01GH002319, OJW), the European Research Council (ERC) under Horizon 2020 research and innovation programme (Grant agreement No. 101003183: A.M.) with additional funding from the Fondazione Romeo & Enrica Invernizzi to the Bocconi Covid Crisis Lab, and a Sir Henry Wellcome Postdoctoral Fellowship (reference 224190/Z/21/Z: C.W.). We are keenly aware of the personal cost of generating the data used in this analysis incurred by both the ZPRIME and ZNPHI teams and would like to acknowledge all researchers and staff who contributed to the collection of data used in these analyses. In particular, we pay tribute to Roy Chavuma, a key public health expert within the team who was lost to COVID-19 in the course of the original mortuary study. We also acknowledge Philip Whiteside, whose journalism during the pandemic, which aimed to provide the public with a better understanding of the global impact of COVID-19, helped to facilitate the early stages of the collaboration underpinning this analysis. The findings and conclusions in this report are those of the authors and do not necessarily represent the official position of the funding agencies or the agencies that the authors are from.

## Author contributions

Authors are represented by their initials as found in the author list. Conceptualization (O.J.W., P.G.T.W.), Methodology (R.J.S., O.J.W., P.G.T.W.), Software (R.J.S., O.J.W., P.G.T.W., N.F.B., L.C.O.), Formal analysis (R.J.S., O.J.W., P.G.T.W.), Investigation (R.J.S., O.J.W., P.G.T.W.), Resources (R.P., J.L., G.K., C.M., N.F.B., S.G., L.C.O., W.M., E.D.F., A.M., J.Z.H., L.B.M., L.M., C.J.G.), Writing—Original Draft (R.J.S., P.G.T.W.), Writing—Review and Editing (all authors), Visualisation (R.J.S., P.G.T.W.), Supervision (P.G.T.W., C.J.G.), Project Administration (P.G.T.W.), Funding Acquisition (P.G.T.W., O.J.W., C.J.G., J.Z.H.).

## Competing interests

S.G. declares shareholdings in pharmaceutical companies (AstraZeneca and GlaxoSmithklineBeecham). L.C.O. declares grant funding for other projects from Merck Ltd. The remaining authors declare no competing interests.
