## [Peer Review File · Nature Communications]

Using mortuary and burial data to place COVID-19 in Lusaka, Zambia within a global contextREVIEWER COMMENTS

Reviewer #1 (Remarks to the Author):

This is an exciting and timely paper that aims to quantify the true impact of the COVID-19 epidemic in African countries. First, the authors use simple calculations to put laboratory-confirmed cases and deaths, estimates of excess deaths, and countries' population structure into context. They propose a demographic vulnerability-weighted impact (DVWI) as a new measure of impact, based on age-specific IFR estimates. They show that presented in this way, the data do not allow clear conclusions about the relative impact across different African countries. Also, it is uncertain whether the impact in the African region was in fact lower than in Europe or the Americas once the population structure is taken into account.

Second, the authors aimed to get more precise estimates for one African setting, Lusaka, Zambia. They use data from burial registrations, seroprevalence data and other data sources to model excess mortality. They conclude that based on their results, it seems unlikely that Zambia and some other African countries have been spared from the impact of the COVID-19 pandemic ("African paradox"). After accounting for the younger age of Lusaka's population, the results indicate that the per capita excess mortality was higher than in South Africa. They argue that "a global inequity in our ability to measure mortality patterns and disease spread in 2020 likely contributed substantially to global inequity in the impact of pandemic itself."

Overall, this paper provides new and interesting insights about an understudied aspect of the COVID-19 pandemic. The methods are original and solid, and the conclusions supported by the data. There are some concerns regarding the many strong assumptions made, and the presentation of the analyses.

Major comments:

1. The paper is not an easy read, and it takes time to fully understand the analyses. Some of the materials in the supplement should be moved to the main text. In particular, a table or graph summarising how the different data sources were used should be included in the main text. Making the paper more accessible to a non-specialist readership will increase its impact.

Points below relate to some strong assumptions made in the analyses. Some of them are mentioned or addressed in sensitivity analyses, but a coherent, comprehensive and accessible discussion addressing the assumptions and the ensuing limitations is lacking.

2. The age-specific IFRs used in the study are based on the study by Brazeau and colleagues, who used seroprevalence data from Europe, Brazil and the US, but no data from Zambia. Further, several of these surveys (for example, in shoppers in New York or blood donors in Switzerland) are unlikely to be representative of the general population. Finally, only confirmed COVID-19 deaths were considered, which will underestimate the true COVID-19-related mortality.

3. The Zambian serosurvey also had limitations, probably underestimating the infection rate (see commentary by Usuf and Roca 2021).

4. The authors assume that the burial registration data covered 90% of all deaths in Lusaka District, but it is unclear what evidence exactly underpins this assumption. The two studies cited relate to a study of BID data ('brought in dead') at the UTH mortuary, which does not report any data on the level of ascertainment, and a verbal autopsy study, which focussed on causes of death. Even if 90% of deaths are usually captured in the burial registry data, this may have been different during the COVID-19 pandemic. As the authors state, 90% is "a belief". The UN data are of limited help in this context - they suffer from the same problem.

5. Following from above, the drop in burial registrations among the under 5 is concerning and not well explained. Why this drop, and why only in the under 5-year-olds? To my eye, Figure 3 indicates that this drop was present in other age groups as well.

6. How applicable to Lusaka are the contact matrix data from Zimbabwe?

7. From a public health and policymaking perspective, the "Demographic vulnerability-weighted

impact" (DVWI), defined as "the level of infection, and corresponding deaths caused directly through COVID-19 infection, necessary to match a given level of excess mortality" is not very intuitive. And how can it be above 100%? Repeat infections? Could it be that the age-specific IFR values used in its calculation come from high-income countries and are thus too low for lower-income countries with limited access to intensive care? In this case, should the DVWI be used to compare African countries?

8. WHO estimates of excess deaths have suffered from important methodological issues in the past (<https://www.nature.com/articles/d41586-022-01526-0>). Did the authors consider using alternative estimates (e.g. Wang et al, 2022 or Karlinsky and Kobak, 2021)

Minor comments:

1. The title mentions "mortality-based surveillance", but didn't the authors use burial registry data?
2. Introduction, line 62: Lines 62-68 are repeated from lines 52-59.
3. Introduction, lines 115-120: This is a discussion of results and implications, I don't think it belongs to the introduction.
4. Results, line 253: I assume these numbers correspond to 887/million from 2020 to mid-2021 and 611/million in 2020? I would reformulate this sentence - difficult to understand. The formulation "during weeks ending in 2020" is also awkward.
5. Results, line 259: Is it correct that the 16.7% refers to the proportion of deaths attributed to COVID-19 in 2020, not to the relative excess?
6. Results, line 303: Both notations R_t and R_{eff} have been used in the literature to refer to the same effective reproduction number, which corresponds to the authors' definition of R_{eff} . To avoid confusion, I would advise the authors to replace R_t by $R_{0,t}$ or $R_0(t)$, as I'm not aware of a precedent.
7. Figure 4: Would it be possible to show the incidence or prevalence by age group over time? I didn't find it in the supplementary file .
8. Results, line 323-355: The reporting of the various sensitivity analyses is interesting but difficult to follow, I would suggest to simplify or maybe add a table summarizing the results of different sensitivity analyses.

Reviewer #2 (Remarks to the Author):

Sheppard et al. use 1) public burial registration data, 2) post-mortem serological test results, 3) prevalence survey data to estimate parameters of a Bayesian SEIR model. Using the estimated parameters, the authors then describe how modelled estimates of COVID-19 severity in Lusaka, Zambia compared to global severity estimates.

The authors find that severity in Lusaka is broadly in agreement with COVID-19 epidemics in other regions of the world and therefore refute the need for exceptional explanations of the low reported disease burden in Zambia (and by extension other countries in SSA). Most of the low reported figures are explained by the young age structure of the population.

The authors rightly highlight the need to address inequities in the capacity to measure mortality and disease spread.

Overall this is a clearly written, if at times overly technical article. The authors report novel and important results that help explain previously paradoxical data on the impact of the COVID-19 pandemic in sub-Saharan Africa (and specifically Lusaka, Zambia for the purpose of this work).

This is important work and methodologically sound. The model is complex, but described to sufficient detail in the supplementary materials and full, clearly annotated code together with the underlying data is made available by the authors on the code repository GitHub.

I do have some comments and queries below, but none of them pertain to serious methodological or interpretational flaws in this study.

CONTEXT

C.1. (l. 71) While reasonably clear from earlier in the introduction, it may not be clear to more casual readers what is meant by the "Africa paradox" - perhaps add a half sentence in brackets to summarise it in a nutshell).

C.2. (l.206-207) Maybe the authors can provide more context for the dip in burial registrations between March & May 2020: is this likely to just be delayed recording or actually fewer deaths driven by the NPIs (e.g. fewer road traffic accidents etc)? This is done somewhat in the discussion but would benefit to be present in the data description and given a bit more space.

C.3. Are there any data on the cause of deaths or is this not captured during burial registrations? If so, this could provide context to some of the data from Figure 2.

METHODS

M.1. The paper would benefit if the authors could give reasons for choosing the time window mid-2017 to mid-2021 (l.100). Why not go back further in time for longer baseline? Why not continue beyond mid-2021 as the authors state themselves that the delta wave was still going on at the time (l.221)? Also why is the analysis eventually limited to Jan 2018 - Jun 2021 (as figures), omitting the data from 2017? OK if this was for convenience given available data, but would be good if clarified.

M.2. I am not fully convinced by the DVWI metric. DVWI estimates the population attack rate required to explain the estimated excess mortality, assuming a specific IFR. So DVWI implicitly assumes all excess mortality is directly due to COVID-19 infections, but there will also be indirect deaths (due to e.g. an overburdened healthcare system, lack of transport to health centres, ...) that the DVWI will attribute to the disease itself. This probably contributes to DVWI exceeding 100% at times (see point I.1.).

M.3. (l.239) The authors are very transparent about the assumptions inherent in using under-5 burial registrations as the reference. I do feel that the manuscript will benefit if the authors would explain their choice for using under 5 burials as the reference a bit more though. Can the authors confirm that the burial registration process (unclear if this is done by the undertakers, the family of the deceased or government / clinical staff) is the same for all burials - if not (e.g. if largely under 5 deaths are registered by maternity wards as opposed to a regional health offices or local undertakers) then this could bias results if the recording of under-5 and over-5 registrations were differentially impacted by the pandemic.

M.4. (l. 256-257) The assumption of a constant and unchanged 'true' under-5 mortality needs more clarification. There is literature showing substantial indirect effects on neonatal outcomes (one example would be <https://doi.org/10.1136/bmjgh-2021-006102>) which would impact under 5 mortality; likewise changes in hospital staffing, procedures and ease of transport to the hospital would impact mortality in any age group. Really what the authors mean here is that there is no reason to assume under-5 mortality would have changed if it were not for the pandemic (excess mortality capturing both direct and indirect effects) and what the authors do here is a comparison of recorded data against a counterfactual scenario.

M.5. (l. 461-466) Were the population structure data assumed to be exactly known census data or was uncertainty from these estimates taken into account? If yes, how?

M.6. (Figure 4 Panel D and Supp Figure 4) There seems to be an outlying post-mortem

prevalence; did the authors investigate that measurement and checked model fit with that value removed?

M.7. (l. 486-7, 514-5 and Supp. Equation (19)) Registration data were assumed to be Poisson-distributed. Was this assumption checked and more flexible count distributions tried?

INTERPRETATION

I.1. (l. 262-268 and Figure 3 H) It might be useful for the reader if the authors could explain briefly how one is supposed to interpret $DWVI > 100\%$? There is nothing stopping $DWVI$ to exceed 100% if other factors drive excess mortality of course, but I was not fully sure how to interpret this in the context here where the implicit assumption is that the excess mortality is driven more or less uniquely by the COVID-19 pandemic. (See also point M.2.)

I.2. (Figure 4 Panels C & F and conclusion l.441-445) I find the drop in modelled seroprevalence and conclusion that the majority of the population remains immune-naive surprising, given serosurveillance data from neighbouring countries (e.g. <https://doi.org/10.1186/s12916-021-02187-y>) that indicate increasing seroprevalence throughout 2020-21. Seroprevalence may be independent from actual immunity of course, but the seroprevalence trends appears contradictory with other data. Maybe the authors can expand on this and what they think are the reasons and implications of rising and falling seroprevalence in Lusaka?

OTHER QUERIES

O.1. (Figure 2) I am confused by why both Lusaka district and province are shown in the top panel. What exactly is the difference and why do they seem to match perfectly Jun 2020 to Feb 2021, then diverge thereafter? Why less data from Lusaka district than province (the latter seems to have data prior to May 2020 and after Mar 2021)?

O.2. A more general comment is that the authors at times mix up SARS-CoV-2 and COVID-19. It is my understanding that SARS-CoV-2 refers to the virus and COVID-19 to the disease caused by SARS-CoV-2 infection as well as the pandemic itself. It is possible to be SARS-CoV-2 infected and asymptomatic, i.e. without COVID-19 disease. So it makes sense to talk about SARS-CoV-2 infection, COVID-19 deaths and COVID-19 pandemic, but, if my understanding is correct, then the authors should amend references to "COVID-19 infections" (examples would be on lines 61, 87, 145). But I realise the literature is not always consistent in the use of these terms.

O.3 Figure 3: Too many panels crammed into a single figure? It is quite hard to make out detail on some of the panels (especially panel B is very difficult). Impossible on a print-out (but admittedly, who is still printing articles?) but even zoomed in on the electronic PDF it is still difficult to make out details. I wonder if e.g. Panel C could be dropped?

O.4. (Figure 4; Panel C) Add a clarification to Panel C that the 2 dots with confidence intervals are the data, the curves the model fit.

O.5. (Figure 4, Panel F) Difficult to see R_t on this graph -- use transparent colours?

O.6. (l. 329) There seems to be a typo here: "0.06-0.125" instead of "0.6-1.25"?

O.7. (Figure 5) Clarify that panel C shows averaged posterior log likelihoods ("posterior model fit" somewhat ambiguous).

Response to Reviewers

We have identified four thematic areas that both reviewers have identified as being key to making the paper publication-ready: i) a focus on making our analysis more digestible for a wider (or even expert) readership; ii) better explanation and justification of our “Demographic vulnerability-weighted impact” metric; iii) a closer examination of the key assumptions made in our analysis of excess mortality; iv) greater clarity and discussion on the various sensitivity analyses we conduct when attempting to assess whether these patterns of excess mortality are consistent with a direct impact of COVID-19.

To ease review, in the following response we have first grouped our responses to reviewer feedback into the four thematic areas above, then provided point-by-point responses to any remaining concerns. We also enclose marked-up and clean versions of both the main manuscript and supplementary information.

As a result of the feedback from the reviewers we believe the manuscript has improved both in terms of helping us to firm up some of our key assumptions but, perhaps even more crucially given the topic, to make the manuscripts more accessible to a wider audience which is something we are hugely grateful to the reviewers for their guidance.

Response to key thematic areas of improvement

i) a focus on making our analysis more digestible for a more general (or even expert) readership

Reviewer 1:

The methods are original and solid, and the conclusions supported by the data. There are some concerns regarding the many strong assumptions made, and the presentation of the analyses...[but]... The paper is not an easy read, and it takes time to fully understand the analyses. Some of the materials in the supplement should be moved to the main text. In particular, a table or graph summarising how the different data sources were used should be included in the main text. Making the paper more accessible to a non-specialist readership will increase its impact.

Reviewer 2:

Overall this is a clearly written, if at times overly technical article.

Our analysis required the development of a bespoke inferential framework to optimise information and insights to be gleaned from combining various datasets showing different snapshots of the epidemic in Lusaka. Providing enough detail about this approach, as well as detail about its assumptions and limitations, whilst maintaining clarity and accessibility for as wide an audience as possible is a difficult task but is critical for the work’s key findings to be understood and believed. Having reviewers with the technical expertise to understand the approach in detail but highlight where it is particularly inaccessibly written has been of huge value to us. We have now substantially rewritten the main text with an eye for clarity. We have broken the results section detailing our excess mortality estimation process into smaller paragraphs, with each paragraph representing each step of the process of transforming burial registration data to an excess mortality estimate for Lusaka, detailing the approach taken and assumptions made (lines 346-399 of marked-up manuscript). We have also provided more context for other more technical aspects of the work – including the justification for the DVWI metric (lines 174-192 of marked-up manuscript) and to attempt to make some of our sensitivity analyses more accessible (e.g. the impact of any shift in age distribution of non-COVID-19 mortality) by giving real-world examples of the kind of shift this would represent (lines 519-544 of marked-up manuscript). We have added a schematic (Figure 1) with a description, in as digestible terms as possible, of our wider inferential approach, and a summary table

(Supplementary Table 1) of the many sensitivity analyses – their rationale and the implications of their results for the key conclusions of analyses.

ii) better explanation and justification of our “Demographic vulnerability-weighted impact” metric

Reviewer 1:

7. From a public health and policymaking perspective, the “Demographic vulnerability-weighted impact” (DVWI), defined as “the level of infection, and corresponding deaths caused directly through COVID-19 infection, necessary to match a given level of excess mortality” is not very intuitive. And how can it be above 100%? Repeat infections? Could it be that the age-specific IFR values used in its calculation come from high-income countries and are thus too low for lower-income countries with limited access to intensive care? In this case, should the DVWI be used to compare African countries?

Reviewer 2:

I am not fully convinced by the DVWI metric. DVWI estimates the population attack rate required to explain the estimated excess mortality, assuming a specific IFR. So DVWI implicitly assumes all excess mortality is directly due to COVID-19 infections, but there will also be indirect deaths (due to e.g. an overburdened healthcare system, lack of transport to health centres, ...) that the DVWI will attribute to the disease itself. This probably contributes to DVWI exceeding 100% at times (see point I.1.).

I.1. (l. 262-268 and Figure 3 H) It might be useful for the reader if the authors could explain briefly how one is supposed to interpret $DVWI > 100\%$? There is nothing stopping DVWI to exceed 100% if other factors drive excess mortality of course, but I was not fully sure how to interpret this in the context here where the implicit assumption is that the excess mortality is driven more or less uniquely by the COVID-19 pandemic. (See also point M.2.)

Our aim with the DVWI index was not to make any assumption, implicit or otherwise, as to what was driving the estimated excess mortality, more simply it was to place the impact in the context of the initial vulnerability of a population to the direct consequences of COVID-19 (i.e., accounting for the likely average severity of an infection). We agree this was not made sufficiently explicit and have now added the text below. On reflection we believe that expressing DVWI as a scalar value rather than a percentage will also help to distinguish that we are not suggesting this value to imply a particular underlying attack rate or a direct causation between infection and impact. Finally, we understand that having values greater than 100% (or 1 in our adjusted metric) can seem counter-intuitive but we believe this a feature rather than an error in that it highlights just how large the estimated impact has been in some countries - i.e., larger than the entire population being infected if assuming a Wuhan- or HIC-based IFR by age. As the reviewers aptly summarise this could be indicating indirect deaths, a higher IFR due to limited access to care (or due to more severe VOC) and/or burden due to reinfection. We have now explained why $DVWI > 1$ can be plausible and used this explanation as a further opportunity to reinforce the point that we are not making any assumption with respect to cause of excess mortality when using this metric:

Lines 174-192 of marked up manuscript (lines 151-166 of revised):

“From these estimates, we then calculated a measure that standardises excess mortality by average protection from (or vulnerability to) severe disease upon infection that comes from population age structure. This measure, the “Demographic vulnerability-weighted impact” (DVWI), is defined explicitly as the cumulative attack rate required to match estimates of excess mortality, assuming direct COVID-19 causation and age-specific IFR from Brazeau et al.⁴², with even spread of infection by age within the population. It is important to note that indirect pandemic consequences also impact excess mortality. Moreover, our estimates are based upon infection-fatality patterns during the pandemic’s first wave, largely using data from high-income settings with good care access and standards, relative to global

averages. Consequently, this measure is not designed to provide insight into excess mortality causes, but places such estimates within the context of the population's vulnerability to direct infection consequences at the beginning of the pandemic. It is, therefore, plausible that countries can have $DVWI > 1$ due to any combination of: i) high indirect pandemic impact; ii) greater disease severity, due to health-care limitations, SARS-CoV-2 variants of greater severity or any other factor not accounted for which contribute to higher IFRs by age than those used in our analysis; iii) substantial burden associated with reinfection."

iii) a closer examination of the key assumptions made in our analysis of excess mortality

Reviewer 1:

The drop in burial registrations among the under 5 is concerning and not well explained. Why this drop, and why only in the under 5-year-olds? To my eye, Figure 3 indicates that this drop was present in other age groups as well.

The authors assume that the burial registration data covered 90% of all deaths in Lusaka District, but it is unclear what evidence exactly underpins this assumption. The two studies cited relate to a study of BID data ('brought in dead') at the UTH mortuary, which does not report any data on the level of ascertainment, and a verbal autopsy study, which focussed on causes of death. Even if 90% of deaths are usually captured in the burial registry data, this may have been different during the COVID-19 pandemic. As the authors state, 90% is "a belief". The UN data are of limited help in this context - they suffer from the same problem.

Reviewer 2:

Maybe the authors can provide more context for the dip in burial registrations between March & May 2020: is this likely to just be delayed recording or actually fewer deaths driven by the NPIs (e.g. fewer road traffic accidents etc)? This is done somewhat in the discussion but would benefit to be present in the data description and given a bit more space.

M.3. (l.239) The authors are very transparent about the assumptions inherent in using under-5 burial registrations as the reference. I do feel that the manuscript will benefit if the authors would explain their choice for using under 5 burials as the reference a bit more though. Can the authors confirm that the burial registration process (unclear if this is done by the undertakers, the family of the deceased or government / clinical staff) is the same for all burials - if not (e.g. if largely under 5 deaths are registered by maternity wards as opposed to a regional health offices or local undertakers) then this could bias results if the recording of under-5 and over-5 registrations were differentially impacted by the pandemic.

M.4. (l. 256-257) The assumption of a constant and unchanged 'true' under-5 mortality needs more clarification. There is literature showing substantial indirect effects on neonatal outcomes (one example would be <https://doi.org/10.1136/bmjgh-2021-006102>) which would impact under 5 mortality; likewise changes in hospital staffing, procedures and ease of transport to the hospital would impact mortality in any age group. Really what the authors mean here is that there is no reason to assume under-5 mortality would have changed if it were not for the pandemic (excess mortality capturing both direct and indirect effects) and what the authors do here is a comparison of recorded data against a counterfactual scenario.

Both reviewers are correct to point out that large declines in registration rates. Declines both during the initial stages of the pandemic and more chronically in periods outside known waves of the pandemic provide a challenge in interpreting the data on burial registration data in terms of excess mortality. Indeed, it is this observation, allied with the observation from reviewer 1, that the age-distribution remained consistent, with rates dipping equally in all ages, during the first acute phase of reduced registration rates

as suppression measures were implemented, which motivated us to look beyond the raw numbers of those registered to the underlying age-patterns within those numbers. As reviewer 2 notes, it is plausible that some decline in non-COVID-19 mortality occurred – e.g., fewer traffic accidents, closure of bars etc. – but very few of these potential reduced risks apply to infants under the age of 5.

All burial registrations occur through the same process involving someone, typically a family member, attending the local permit office, and are required for all ages. Registration does involve different forms for those that occur inside and outside of hospital so it is perhaps conceivable that registration of age groups more likely to die in hospital (e.g. neonates) were differentially disrupted than those outside of hospital. However, similarly to the logic above, it is difficult to reconcile any such potential effect with the similar patterns in all ages during the initial lockdown phase, followed by the similar trends in infant, young children and young adults throughout the study period in general.

We have now provided more context around the consistent trends across all age groups in which SARS-CoV-2 infection is typically considered more likely to cause mild COVID-19 symptoms, this involves presenting the synchronized trends more clearly in Figure 3C and highlighting that these occur despite mortality in these different age groups having quite diverging aetiologies. Moreover, we have also conducted a parallel analysis using 5-15 as our reference age-group to guide excess mortality trends in older ages, demonstrating this produces very similar results to the under 5 category (see Supplementary Figure 3).

We believe the most parsimonious explanation, and therefore our default hypothesis, for the consistent drop in these younger age categories, and across all ages when known waves of SARS-CoV-2 infection are excluded (e.g March-May 2020 during lockdown, Oct-Dec 2020 prior to Beta wave, April-May 2021 between Beta and Delta waves) is that, as for many aspects of life, there was general disruption to burial registration processes throughout the pandemic, either due to reductions in the likelihood of family members seeking registration certificates or registrations being recorded. However, in the absence of definitive proof that this occurred we cannot rule out more complicated alternative hypotheses that could also generate such correlations in patterns of registrations by age. In light of this, we also prominently present estimates without any scaling (now more clearly presented in Table 1) and conduct multiple supplementary analyses looking at the influence shifts in age distribution of non-COVID-19 deaths during the pandemic could have on our key results (see Supplementary Table 1).

It should be noted that our assumption that reductions in under 5 registrations can be used as guide to disruption in non-COVID-19 burial registration rates will only over-estimate excess mortality in the event that under 5 mortality has actually *reduced* during the pandemic relative to pre-pandemic levels. As reviewer 2 very helpfully notes, in other settings an increase in neonatal mortality has been observed, if under 5 mortality had indeed increased during the pandemic, our assumption could therefore be underestimating the wider level of disruption to registration services and therefore represent a conservative estimate of excess mortality in the older age groups. This factor has now been highlighted in the discussion (lines 599-601 of marked up manuscript), alongside the very useful reference which we thank the reviewer for bringing to our attention.

Finally, as reviewer 1 notes, our use of 90% mortality registration represents a consensus of local experts in the field. We have subsequently added to this consensus by consulting with Dr Stephen Chanda, mortality surveillance coordinator at the Zambia National Public Health Institute, who has reassured us that whilst this belief appears reasonable there are no concrete data we can use to validate it. Meanwhile, our initial impulse to argue that UN data at least represent an alternative source that take advantage of non-routine sources such as census data and population-based surveys, our enthusiasm to make such an

argument has been somewhat dampened by the fact that these estimates appear to have recently changed in the UN's latest update to their population projections! Instead, we have now provided the reader with more information with respect to the implications of this assumption. Firstly, that we can provide a hard limit to the extent to which this assumption can contribute to an overstating of excess mortality, with a situation where 100% of deaths are registered providing the most conservative assumption possible. We also provide an alternative scenario where the proportion of deaths that go unregistered is doubled (i.e. an 80% capture rate) producing a higher number of excess deaths (Table 1). We then make the point that any higher proportion of unregistered deaths would lead to concomitantly higher estimates of excess deaths, in turn strengthening the overarching finding of the work that Lusaka was in no way 'spared' by the pandemic (see lines 384-395 of marked up manuscript).

iv) more clarity and discussion on the various sensitivity analyses we conduct when attempting to assess whether these patterns of excess mortality are consistent with a direct impact of COVID-19

Reviewer 1:

8. Results, line 323-355: The reporting of the various sensitivity analyses is interesting but difficult to follow, I would suggest to simplify or maybe add a table summarizing the results of different sensitivity analyses.

Reviewer 2:

M.6. (Figure 4 Panel D and Supp Figure 4) There seems to be an outlying post-mortem prevalence; did the authors investigate that measurement and checked model fit with that value removed?

We agree that in our attempts to provide as thorough an analysis of the implications of our key assumptions we have generated a slightly unwieldy number of results that are difficult to follow. The suggestion of a table to summarise these analyses is an excellent one that we have now added to the Supplementary Information (see Supplementary Table 1). We have also attempted to better subset out analyses in terms of those looking to assess the influence of outliers, (as with the one raised by reviewer 2 which is represented "B" in Supplementary Table 1 and Supplementary Figure 6), those involving assumptions around model parameters, and those involving assumption around underlying patterns of non-COVID-19 mortality.

REVIEWER COMMENTS – full and addressed:

Reviewer #1 (Remarks to the Author):

This is an exciting and timely paper that aims to quantify the true impact of the COVID-19 epidemic in African countries. First, the authors use simple calculations to put laboratory-confirmed cases and deaths, estimates of excess deaths, and countries' population structure into context. They propose a demographic vulnerability-weighted impact (DVWI) as a new measure of impact, based on age-specific IFR estimates. They show that presented in this way, the data do not allow clear conclusions about the relative impact across different African countries. Also, it is uncertain whether the impact in the African region was in fact lower than in Europe or the Americas once the population structure is taken into account.

Second, the authors aimed to get more precise estimates for one African setting, Lusaka, Zambia. They use data from burial registrations, seroprevalence data and other data sources to model excess mortality. They conclude that based on their results, it seems unlikely that Zambia and some other African countries have been spared from the impact of the COVID-19 pandemic ("African paradox"). After accounting for the younger age of Lusaka's population, the results indicate that the per capita excess mortality was higher than in South Africa. They argue that "a global inequity in our ability to measure mortality patterns and disease spread in 2020 likely contributed substantially to global inequity in the impact of pandemic itself."

Overall, this paper provides new and interesting insights about an understudied aspect of the COVID-19 pandemic. The methods are original and solid, and the conclusions supported by the data. There are some concerns regarding the many strong assumptions made, and the presentation of the analyses.

We thank the reviewer for their positive and in-depth review and highly constructive criticism.

Major comments:

1. The paper is not an easy read, and it takes time to fully understand the analyses. Some of the materials in the supplement should be moved to the main text. In particular, a table or graph summarising how the different data sources were used should be included in the main text. Making the paper more accessible to a non-specialist readership will increase its impact.

Please see above - major theme i)

Points below relate to some strong assumptions made in the analyses. Some of them are mentioned or addressed in sensitivity analyses, but a coherent, comprehensive and accessible discussion addressing the assumptions and the ensuing limitations is lacking.

Please see above - major theme iii)

2. The age-specific IFRs used in the study are based on the study by Brazeau and colleagues, who used seroprevalence data from Europe, Brazil and the US, but no data from Zambia. Further, several of these surveys (for example, in shoppers in New York or blood donors in Switzerland) are unlikely to be representative of the general population. Finally, only confirmed COVID-19 deaths were considered, which will underestimate the true COVID-19-related mortality.

We have added additional discussion on our default IFR assumptions and how they relate to comparable estimates (lines 643-663 of marked up manuscript, 482-501 of revised):

“Our default IFR assumptions are based on results from Brazeau et al.⁴², calculated using data matching our main study period time-frame (i.e., prior to the emergence of new variants of possible differing severity). As with all severity studies during the initial pandemic stages, this study included data representing a trade-off between study-design quality and the representation of a wide range of contexts, leading to data inclusion with some potential measurement error. Bias may therefore be present in some population exposure data (i.e., data collected through convenience sampling including shopping centre attendees and blood donors) and COVID-19 mortality data (i.e., where confirmed COVID-19 mortality use can underestimate total attributable mortality). However, the included data come from countries with strong testing systems, and crude IFR estimates from convenience-sample sources are not dissimilar to other included estimates. Other studies have suggested a higher IFR for very young children relative to older children⁶⁷⁻⁶⁹, which might account for the high observed U5 post-mortem prevalence, though at levels (<0.01%) that would make negligible difference to the fit of our model to the data. A plausible explanation for this U5 prevalence, though not one quantifiable in our framework, could be comparatively extensive SARS-CoV-2 spread within communities of high non-COVID-19-associated infant mortality. Overall, Brazeau et al. IFR estimates are central within the range of other estimates⁴². Thus, it seems plausible that an ensemble approach could broaden our uncertainty but would be unlikely to alter our central conclusion that, when analysing one of the best-characterised epidemics in sub-Saharan Africa, there is no evidence to support any substantial differences between innate COVID-19 severity in Lusaka relative to estimates from other parts of the world.

3. The Zambian serosurvey also had limitations, probably underestimating the infection rate (see commentary by Usuf and Roca 2021).

Usuf and Roca (2021) state that “... only half of the participants agreed to provide samples for both PCR and ELISA and, therefore, overall infection rate is probably higher than reported in the survey.”

In our analysis we do not restrict our estimates to individuals who provide both samples so our response rate is substantially higher (around 70% for both metrics). As with any study, that we are not able to use data from all individuals involved in the study could introduce bias. However, as we are using the proportion of positive samples in those for whom there was a sample, this will only be if those for whom we do not have a result have systematically higher or lower prevalence than those for whom we do (a limitation of any survey with any rate of refusal). In the absence of any systematic difference our results would be unbiased but, given a lower sample size, be subject to wider uncertainty, a factor accounted for in our binomial likelihood. Other factors like specimen storage and analytic sensitivity of the assay could impact the estimates, but these are well-acknowledged limitations of SARS-CoV-2 prevalence studies. We also do attempt to account for the sensitivity of the assay as a function of time from exposure within our estimation process (See Supplementary Figure 11).

4. The authors assume that the burial registration data covered 90% of all deaths in Lusaka District, but it is unclear what evidence exactly underpins this assumption. The two studies cited relate to a study of BID data ('brought in dead') at the UTH mortuary, which does not report any data on the level of ascertainment, and a verbal autopsy study, which focussed on causes of death. Even if 90% of deaths are usually captured in the burial registry data, this may have been different during the COVID-19 pandemic. As the authors state, 90% is "a belief". The UN data are of limited help in this context - they suffer from the same problem.

Please see above – major theme iii)

5. *Following from above, the drop in burial registrations among the under 5 is concerning and not well explained. Why this drop, and why only in the under 5-year-olds? To my eye, Figure 3 indicates that this drop was present in other age groups as well.*

Please see above – major theme iii)

6. *How applicable to Lusaka are the contact matrix data from Zimbabwe?*

We have now addressed this in the main text as below (lines 754-763 of marked up manuscript) – in the context of this study the main feature of an adequate contact matrix is that it replicates patterns of exposure by age. As a result, in terms of validation we have also provided a comparison between the age distribution of attack rates generated by this matrix and the age distribution within the population-based survey, acknowledging the small sample size this leads to within the data (Supplementary Figure 5).

Lines 754-763 of marked up manuscript (580-588 revised manuscript):

"In the absence of locally collected data on social contact patterns we used a social contact matrix obtained from social contact data collected from Manicaland, Zimbabwe, the nearest geographical location in the literature, filtering the data to only include data from the peri-urban region (Nyanga) within the dataset⁸⁶. As is generally the case in data collected from Lower-Income countries⁸⁷, this matrix produces attack rates throughout an epidemic which are much flatter by age than any equivalent simulation using data from Higher Income settings. For validation, PCR prevalence and seroprevalence patterns by age as observed in the population-based survey in Lusaka were compared with those estimated contemporaneously by the model."

7. *From a public health and policymaking perspective, the "Demographic vulnerability-weighted impact" (DVWI), defined as "the level of infection, and corresponding deaths caused directly through COVID-19 infection, necessary to match a given level of excess mortality" is not very intuitive. And how can it be above 100%? Repeat infections? Could it be that the age-specific IFR values used in its calculation come from high-income countries and are thus too low for lower-income countries with limited access to intensive care? In this case, should the DVWI be used to compare African countries?*

Please see above – major theme ii)

8. *WHO estimates of excess deaths have suffered from important methodological issues in the past (<https://www.nature.com/articles/d41586-022-01526-0>). Did the authors consider using alternative estimates (e.g. Wang et al, 2022 or Karlinsky and Kobak, 2021)*

It could be argued that much of the focus on WHO estimates has been somewhat Eurocentric and upon aspects which are relatively nuanced when considered at a global level. That said, there are estimates in Africa that are equally, if not more, difficult to reconcile – for example Togo and Kenya where WHO suggests there are 'partial data' upon which they can be effectively certain that there was negative excess mortality in 2020 but do not provide a source for this data nor can we find any evidence in the literature that this was the case. In our opinion, however, these are the most high-profile estimates of excess mortality available (as evidenced by the source the reviewer cites highlighting the large consequences of their potential flaws) and as such we believe are the best to use to help highlight to the reader that there are no widely accepted quantitative basis to base any preconception that excess mortality in Africa was unexpectedly low during the pandemic. Of the cited alternative estimates Wang et al. do not disaggregate to 2020 so do not admit direct comparison but similarly have no direct data on all-cause mortality for any country in sub-Saharan Africa outside of South Africa upon which to base any estimation. Meanwhile, Karlinsky and Kobak do not show estimates for the majority of sub-Saharan countries.

Minor comments:

1. The title mentions “mortuary-based surveillance”, but didn't the authors use burial registry data?

We used both burial registry data and results from mortuary-based PCR sampling. We have changed the title to be more inclusive of these data sources:

“Using mortuary and burial data to place COVID-19 in Lusaka, Zambia within a global context”

2. Introduction, line 62: Lines 62-68 are repeated from lines 52-59.

This has now been amended. We thank the reviewer for bringing this to our attention.

3. Introduction, lines 115-120: This is a discussion of results and implications, I don't think it belongs to the introduction.

Though we would be very happy to remove this, it was originally added as part of a requirement of the journals manuscript checklist (Nature Communications checklist). We leave this to the journal's editorial team to confirm.

4. Results, line 253: I assume these numbers correspond to 887/million from 2020 to mid-2021 and 611/million in 2020? I would reformulate this sentence - difficult to understand. The formulation "during weeks ending in 2020" is also awkward.

This sentence has now been reformulated:

Lines 357-360 of marked up manuscript (281-283 of revised manuscript):

‘These estimates correspond with 644.1 (95% CrI: 471.6-810.7) and 909.7 (95% CrI: 670.6-1,140.7) excess registrations per million total population during 2020 and during January 2020-June 2021 respectively.’

5. Results, line 259: Is it correct that the 16.7% refers to the proportion of deaths attributed to COVID-19 in 2020, not to the relative excess?

This estimate has been replaced with an estimate of excess mortality relative to median 2018-2019 burial registrations, with a corresponding estimate for excess burial registrations. We believe these estimates refer to similar quantities without being so convoluted:

Lines 360-362 of marked up manuscript (284-285 of revised manuscript):

“The estimates also represent 10.3% (95% CrI: 7.6-12.9%) and 10.5% (95% CrI: 7.7-13.3%) of median pre-pandemic burial registrations during 2020-June 2021 and 2020, respectively (Table 1)”

395-399 of marked up manuscript (306-310 of revised manuscript)

“Our excess death estimates represented 18.5% (95% CrI: 13.0-25.2%) and 17.6% (95% CrI: 13.0-23.0%) of pre-pandemic burial registrations for the two respective time-periods, exceeding 50% of 2018-2019 median registrations (Supplementary Figure 4, approaching 150% when filtered to deaths over 50 years) during the peaks of all three waves, are robust to this registration coverage uncertainty.”

6. *Results, line 303: Both notations R_t and R_{eff} have been used in the literature to refer to the same effective reproduction number, which corresponds to the authors' definition of R_{eff} . To avoid confusion, I would advise the authors to replace R_t by $R_{\{0,t\}}$ or $R_0(t)$, as I'm not aware of a precedent.*

We have replaced R_t with $R_0(t)$ as recommended.

7. *Figure 4: Would it be possible to show the incidence or prevalence by age group over time? I didn't find it in the supplementary file.*

We have now included a figure that shows population-level PCR prevalence and seroprevalence through time by age group, comparing this to the available data from the population-based survey (see Supplementary Figure 5).

8. *Results, line 323-355: The reporting of the various sensitivity analyses is interesting but difficult to follow, I would suggest to simplify or maybe add a table summarizing the results of different sensitivity analyses.*

Please see above – major theme iv)

Reviewer #2 (Remarks to the Author):

Sheppard et al. use 1) public burial registration data, 2) post-mortem serological test results, 3) prevalence survey data to estimate parameters of a Bayesian SEIR model. Using the estimated parameters, the authors then describe how modelled estimates of COVID-19 severity in Lusaka, Zambia compared to global severity estimates.

The authors find that severity in Lusaka is broadly in agreement with COVID-19 epidemics in other regions of the world and therefore refute the need for exceptional explanations of the low reported disease burden in Zambia (and by extension other countries in SSA). Most of the low reported figures are explained by the young age structure of the population.

The authors rightly highlight the need to address inequities in the capacity to measure mortality and disease spread.

Overall this is a clearly written, if at times overly technical article. The authors report novel and important results that help explain previously paradoxical data on the impact of the COVID-19 pandemic in sub-Saharan Africa (and specifically Lusaka, Zambia for the purpose of this work). This is important work and methodologically sound. The model is complex, but described to sufficient detail in the supplementary materials and full, clearly annotated code together with the underlying data is made available by the authors on the code repository GitHub.

I do have some comments and queries below, but none of them pertain to serious methodological or interpretational flaws in this study.

We thank the reviewer for their positive feedback and hope they can see how much the work has benefitted from their input.

CONTEXT

C.1. (l. 71) While reasonably clear from earlier in the introduction, it may not be clear to more casual readers what is meant by the "Africa paradox" - perhaps add a half sentence in brackets to summarise it in a nutshell).

We have now added the following to clarify this point:

Lines 77-81 of marked up manuscript (63-67 of revised manuscript)]

“Understanding whether there was any so-called “Africa paradox”^{12,16,17} (the perception of lower COVID-19 impacts across many African countries compared with expectations) and if so, which, if any, of these hypotheses have justifiable basis can help ensure that the correct conclusions and lessons are learned from the pandemic in Africa.”

C.2. (l.206-207) Maybe the authors can provide more context for the dip in burial registrations between March & May 2020: is this likely to just be delayed recording or actually fewer deaths driven by the NPIs (e.g. fewer road traffic accidents etc)? This is done somewhat in the discussion but would benefit to be present in the data description and given a bit more space.

See above – major theme iii)

C.3. Are there any data on the cause of deaths or is this not captured during burial registrations? If so, this could provide context to some of the data from Figure 2.

The cause of death was not captured in the dataset we have access to, but we agree it would be extremely useful and interesting – at least to separate out accidental deaths that may have been affected from NPIs.

METHODS

M.1. The paper would benefit if the authors could give reasons for choosing the time window mid-2017 to mid-2021 (l.100). Why not go back further in time for longer baseline? Why not continue beyond mid-2021 as the authors state themselves that the delta wave was still going on at the time (l.221)? Also why is the analysis eventually limited to Jan 2018 - Jun 2021 (as figures), omitting the data from 2017? OK if this was for convenience given available data, but would be good if clarified.

Our burial registration dataset begins in mid-2017 so we were not able to go back further than that year. Furthermore, the data prior to 2018 was somewhat volatile (perhaps as registration processes ramped up), becoming more consistent from 2018. The dataset also ends in June 2021 and while we are in discussions with collaborators to extend the excess mortality analysis further, this is outside the scope of this analysis, which is primarily focused on COVID-19 impact and severity during the first phase of the pandemic. The following has been added to the text to explain this:

Lines 722-725 of marked up manuscript (552-554 of revised manuscript):

“Although the burial registration dataset begins in 2017, we found that registrations in 2017 increase from low numbers, only reaching greater stability and reliability from 2018, and therefore censored registrations during 2017 in our analysis.”

M.2. I am not fully convinced by the DVWI metric. DVWI estimates the population attack rate required to explain the estimated excess mortality, assuming a specific IFR. So DVWI implicitly assumes all excess mortality is directly due to COVID-19 infections, but there will also be indirect deaths (due to e.g. an overburdened healthcare system, lack of transport to health centres, ...) that the DVWI will attribute to the disease itself. This probably contributes to DVWI exceeding 100% at times (see point I.1.).

See above – major theme ii)

M.3. (l.239) The authors are very transparent about the assumptions inherent in using under-5 burial registrations as the reference. I do feel that the manuscript will benefit if the authors would explain their choice for using under 5 burials as the reference a bit more though. Can the authors confirm that the burial registration process (unclear if this is done by the undertakers, the family of the deceased or government / clinical staff) is the same for all burials - if not (e.g. if largely under 5 deaths are registered by maternity wards as opposed to a regional health offices or local undertakers) then this could bias results if the recording of under-5 and over-5 registrations were differentially impacted by the pandemic.

See above – major theme iii)

M.4. (l. 256-257) The assumption of a constant and unchanged 'true' under-5 mortality needs more clarification. There is literature showing substantial indirect effects on neonatal outcomes (one example would be <https://doi.org/10.1136/bmjgh-2021-006102>) which would impact under 5 mortality; likewise changes in hospital staffing, procedures and ease of transport to the hospital would impact mortality in any age group. Really what the authors mean here is that there is no reason to assume under-5 mortality would have changed if it were not for the pandemic (excess mortality capturing both direct and indirect effects) and what the authors do here is a comparison of recorded data against a counterfactual scenario.

See above – major theme iii)

M.5. (l. 461-466) Were the population structure data assumed to be exactly known census data or was uncertainty from these estimates taken into account? If yes, how?

These estimates are taken from UN World Population Prospects projections of population by age. Though as noted above, the imperfect nature of these projections can be observed across different revisions, they do attempt to extrapolate from various data sources including census data and population-based surveys where available. To our knowledge there is no readily available means by which to incorporate the underlying uncertainty in these projections, however, as these represent the official United Nations population estimates, and reflect well understood demographic trends in that population in sub-Saharan Africa, and Zambia in particular, are very young relative to global averages, we do not anticipate this particular aspect of our analysis to be amongst the most questioned by the wider scientific community relative to the other aspects of our analysis. Given the wide range of sensitivity analyses conducted, and uncertainty captured, from our other inputs and assumptions, which have already been noted by both reviewers to be thorough but hard to digest, we have opted to not explore this factor further in our current analysis.

M.6. (Figure 4 Panel D and Supp Figure 4) There seems to be an outlying post-mortem prevalence; did the authors investigate that measurement and checked model fit with that value removed?

See above – major theme iv)

M.7. (l. 486-7, 514-5 and Supp. Equation (19)) Registration data were assumed to be Poisson-distributed. Was this assumption checked and more flexible count distributions tried?

We have now tested this model fitting assuming a negative binomial distribution under a range of dispersion parameters. We find that the log likelihood of the fit improves as the overdispersion decreases and that the Poisson distribution produced a better fit to the data than the negative binomial for any of our overdispersion parameters tested (see Supplementary Figure 9).

INTERPRETATION

I.1. (l. 262-268 and Figure 3 H) It might be useful for the reader if the authors could explain briefly how one is supposed to interpret $DWVI > 100\%$? There is nothing stopping $DWVI$ to exceed 100% if other factors drive excess mortality of course, but I was not fully sure how to interpret this in the context here where the implicit assumption is that the excess mortality is driven more or less uniquely by the COVID-19 pandemic. (See also point M.2.)

See above – major theme ii)

I.2. (Figure 4 Panels C & F and conclusion l.441-445) I find the drop in modelled seroprevalence and conclusion that the majority of the population remains immune-naive surprising, given serosurveillance data from neighbouring countries (e.g. <https://doi.org/10.1186/s12916-021-02187-y>) that indicate increasing seroprevalence throughout 2020-21. Seroprevalence may be independent from actual immunity of course, but the seroprevalence trends appears contradictory with other data. Maybe the authors can expand on this and what they think are the reasons and implications of rising and falling seroprevalence in Lusaka?

Editorial Note: Below right figure reproduced from Mandolo, J., Msefula, J., Henrion, M.Y.R. *et al.* SARS-CoV-2 exposure in Malawian blood donors: an analysis of seroprevalence and variant dynamics between January 2020 and July 2021. *BMC Med* **19**, 303 (2021). <https://doi.org/10.1186/s12916-021-02187-y> (CC BY 4.0)

We believe a small decline in seroprevalence around three months after the epidemic peak in our estimates is to be expected given known average time to sero-reversion of around 100 days (data based on Muecksch *et al.* (2020), Supplementary Figure 11), however we do also track the cumulative attack rate in our model. This produces a somewhat smaller proportion of immune-naïve individuals, though still the majority of the population. When matching by study period we are very happy to see that our estimates (below left- noting time-period is in 2020) are actually very similar to the Malawian data cited by the reviewer (below right) which also show seroprevalence peaking at a maximum of around 20% and declining around October 2020. We also have seen unpublished data for an area of rural Mozambique showing similar results so are confident our results will not prove to be anomalous for the region. We are hesitant to assume an IFR for the subsequent beta and delta waves in order to extrapolate attack rates throughout 2021 but we believe patterns of excess mortality in our analysis would be highly likely to be consistent with the majority of the population sero-converting throughout these waves. This would again be highly consistent with both the Malawian data and the unpublished Mozambique data. We have now cited the Malawian data to support the generalizability of our finding and thank the reviewer for bringing this reference to our attention.

Lines 668-671 of marked up manuscript (506-508 of revised manuscript):

“Following this initial exponential growth period, our estimates suggest that control of the epidemic was re-established around August 2020, showing patterns that closely resemble trends in Malawian prevalence data⁷⁶.”

OTHER QUERIES

O.1. (Figure 2) I am confused by why both Lusaka district and province are shown in the top panel. What exactly is the difference and why do they seem to match perfectly Jun 2020 to Feb 2021, then diverge thereafter? Why less data from Lusaka district than province (the latter seems to have data prior to May 2020 and after Mar 2021)?

This figure previously represented data scraped from a number of irregularly reported sources, some of which no longer appear to be publicly available. We have now reverted to data from a now publicly available dashboard for Lusaka province produced by the National Public Health Institute in collaboration with the Zambian Ministry of Health ([Zambia \(COVID-19\) General Dashboard \(arcgis.com\)](https://arcgis.com)).

O.2. A more general comment is that the authors at times mix up SARS-CoV-2 and COVID-19. It is my understanding that SARS-CoV-2 refers to the virus and COVID-19 to the disease caused by SARS-CoV-2 infection as well as the pandemic itself. It is possible to be SARS-CoV-2 infected and asymptomatic, i.e. without COVID-19 disease. So it makes sense to talk about SARS-CoV-2 infection, COVID-19 deaths and COVID-19 pandemic, but, if my understanding is correct, then the authors should amend references to "COVID-19 infections" (examples would be on lines 61, 87, 145). But I realise the literature is not always consistent in the use of these terms.

We have now made the recommended changes for consistency.

O.3 Figure 3: Too many panels crammed into a single figure? It is quite hard to make out detail on some of the panels (especially panel B is very difficult). Impossible on a print-out (but admittedly, who is still printing articles?) but even zoomed in on the electronic PDF it is still difficult to make out details. I wonder if e.g. Panel C could be dropped?

We have removed panel C as suggested.

O.4. (Figure 4; Panel C) Add a clarification to Panel C that the 2 dots with confidence intervals are the data, the curves the model fit.

We have separated the data from the model in the legend of this figure panel.

O.5. (Figure 4, Panel F) Difficult to see R_t on this graph -- use transparent colours?

We have made these curves transparent in the plot as suggested.

O.6. (l. 329) There seems to be a typo here: "0.06-0.125" instead of "0.6-1.25"?

This has been corrected in the text.

O.7. (Figure 5) Clarify that panel C shows averaged posterior log likelihoods ("posterior model fit" somewhat ambiguous).

We have now clarified that this represents the log of the averaged posterior likelihoods.

** See Nature Portfolio's author and referees' website at www.nature.com/authors for information about policies, services and author benefits.

REVIEWERS' COMMENTS

Reviewer #1 (Remarks to the Author):

Thank you very much for the thoughtful revision, which addressed all our concerns.

Reviewer #2 (Remarks to the Author):

I have no further comments. The authors have comprehensively addressed my previous comments and I wish to congratulate them on a beautiful and important article.

I would urge the authors though to make all underlying data available through the GitHub (or other repository). I understand this may be difficult for data owned by the Zambian government. Most the data underlying the work in this paper are only needed as aggregate data without any data privacy issues and so I do not see why there needs to be gatekeeping via access requests as outlined in the data availability statement.

REVIEWERS' COMMENTS

Reviewer #1 (Remarks to the Author):

Thank you very much for the thoughtful revision, which addressed all our concerns.

Reviewer #2 (Remarks to the Author):

I have no further comments. The authors have comprehensively addressed my previous comments and I wish to congratulate them on a beautiful and important article.

I would urge the authors though to make all underlying data available through the GitHub (or other repository). I understand this may be difficult for data owned by the Zambian government. Most the data underlying the work in this paper are only needed as aggregate data without any data privacy issues and so I do not see why there needs to be gatekeeping via access requests as outlined in the data availability statement.

Following discussion with co-authors, we have now included substantial data with the repository, including IFR estimates used to generate figure 2, the Nyanga social contact matrix, and aggregated burial registration, post-mortem PCR prevalence, and population PCR prevalence and seroprevalence data. The only data not given in the data are population demography estimates for Lusaka that have not been published elsewhere (similar estimates can found online, e.g. from [Zambia Statistics Agency – Quality Statistics for Development \(zamstats.gov.zm\)](https://zamstats.gov.zm)). We hope these will be sufficient for a robust investigation of our analyses, if desired by readers.